# High-order sensory processing nanocircuit based on coupled VO$_2$ oscillators

Ke Yang[1,7], Yanghao Wang [1,7], Pek Jun Tiw[1], Chaoming Wang[2], Xiaolong Zou[2], Rui Yuan[1], Chang Liu[1], Ge Li[3], Chen Ge [3], Si Wu[2], Teng Zhang [1] ✉, Ru Huang [1] & Yuchao Yang [1,4,5,6] ✉

Conventional circuit elements are constrained by limitations in area and power efficiency at processing physical signals. Recently, researchers have delved into high-order dynamics and coupled oscillation dynamics utilizing Mott devices, revealing potent nonlinear computing capabilities. However, the intricate yet manageable population dynamics of multiple artificial sensory neurons with spatiotemporal coupling remain unexplored. Here, we present an experimental hardware demonstration featuring a capacitance-coupled VO$_2$ phase-change oscillatory network. This network serves as a continuous-time dynamic system for sensory pre-processing and encodes information in phase differences. Besides, a decision-making module for special post-processing through software simulation is designed to complete a bio-inspired dynamic sensory system. Our experiments provide compelling evidence that this transistor-free coupling network excels in sensory processing tasks such as touch recognition and gesture recognition, achieving significant advantages of fewer devices and lower energy-delay-product compared to conventional methods. This work paves the way towards an efficient and compact neuromorphic sensory system based on nano-scale nonlinear dynamics.

The emergence of technological innovations, such as wearable electronics[1], auto-driving[2], and virtual reality[3], are calling for advanced sensory systems, that minimize redundant data movement between sensors and processing units, thereby enhancing area and energy efficiencies. In a typical sensory system, the sensory data follows a hierarchical processing flow, encompassing low-level sensory processing (e.g., encoding, filtering, and feature enhancement) to high-level abstract representation (e.g., recognition, classification, and localization)[4]. The biological sensory system adopts an efficient method to handle sensing in the noisy analog domain. Signals pass through skin receptors and afferent neurons for pre-processing and finally reach to spinal cord for post-processing (Fig. 1a).

With regards to low-level sensory processing, it's vital to encode and transmit the proliferated data from the sensory nodes efficiently. However, the conventional architecture suffers from inefficient power consumption and notable latency. As shown in Fig. 1b, the detected data from the sensors must initially undergo digitization through an analog-to-digital (ADC) circuit before being temporarily stored in a memory unit, awaiting processing by the computing unit. This process introduces significant time delays and energy consumption. To address this issue, novel computing primitives such as near-sensor computing and in-sensor computing have been proposed and demonstrated on emerging nanoelectronics devices[5–7]. In near-sensor computing, processing units or accelerators are located beside sensors

[1]Beijing Advanced Innovation Center for Integrated Circuits, School of Integrated Circuits, Peking University, Beijing 100871, China. [2]School of Psychological and Cognitive Sciences, IDG/McGovern Institute for Brain Research, PKU-Tsinghua Center for Life Sciences, Peking University, Beijing 100871, China. [3]Beijing National Laboratory for Condensed Matter Physics, Institute of Physics, Chinese Academy of Sciences, Beijing 100190, China. [4]Center for Brain Inspired Chips, Institute for Artificial Intelligence, Frontiers Science Center for Nano-optoelectronics, Peking University, Beijing 100871, China. [5]School of Electronic and Computer Engineering, Peking University, Shenzhen 518055, China. [6]Center for Brain Inspired Intelligence, Chinese Institute for Brain Research (CIBR), Beijing 102206, China. [7]These authors contributed equally: Ke Yang, Yanghao Wang. ✉e-mail: tengzhang@pku.edu.cn; yuchaoyang@pku.edu.cn

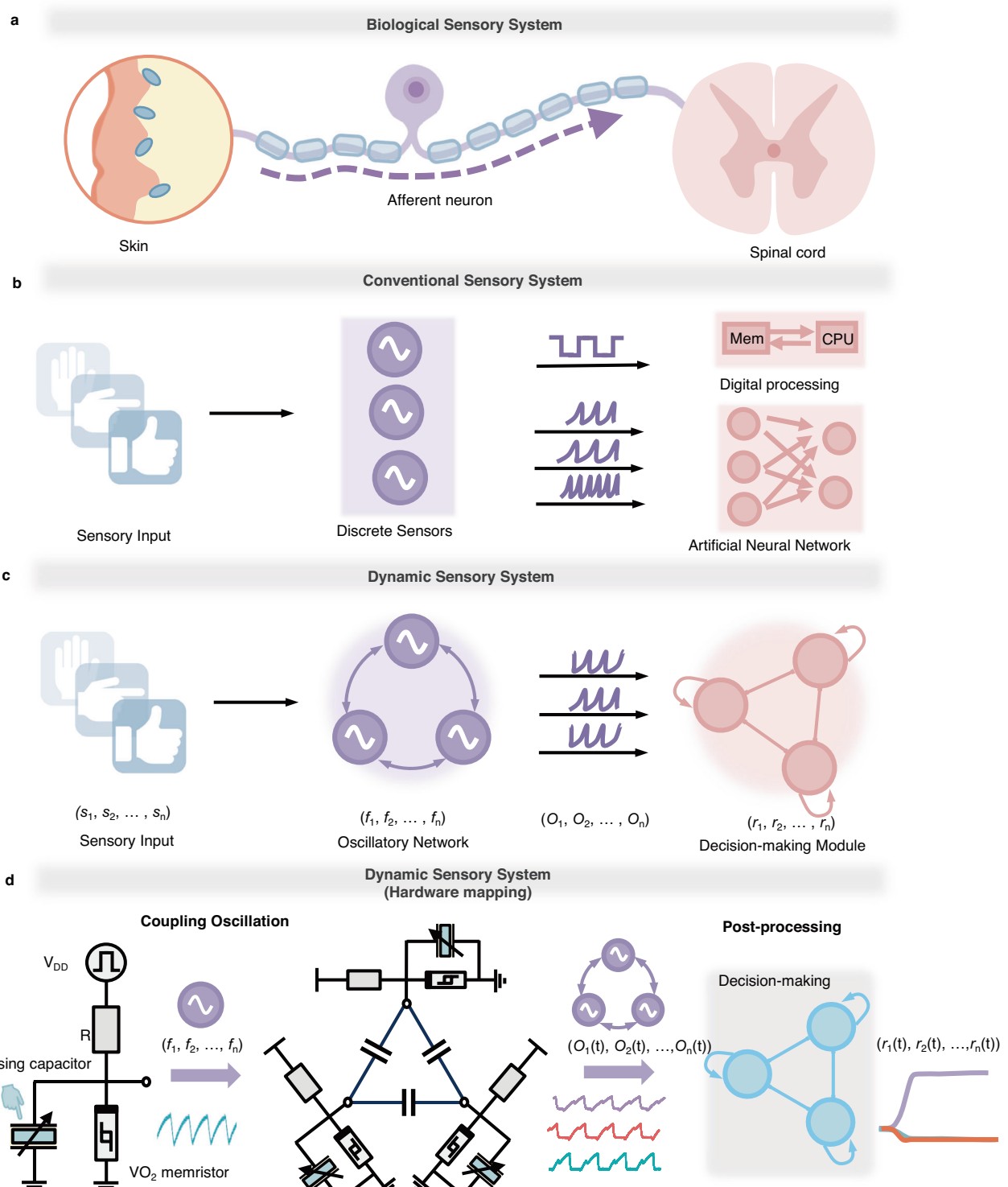

**Fig. 1 | The sensory processing framework in different systems. a** Biological sensory system. Receptors and neurons convert physical stimuli into electrical pulses which are transmitted to the cortex for high-level processing. **b** Conventional sensory system. In digital processing way, the signals from sensors are first digitalized through ADC and then put into separate memory units and processing units. In the ANN way, a crossbar composed of sensors can perform linear operations and realize classification. **c** Dynamic sensory system. The proposed sensory system consists of an oscillatory network and a decision-making module. When receiving a sensory mode **s**, the coupling dynamic will convert discrete oscillator frequencies f into phase differences O at the same frequency through coupling. Then the decision-making module outputs a classification result r. **d** Hardware mapping of the dynamic sensory system. The circuit can encode the sensory information into the phase pattern O after electrical coupling. The post-processing of the decision-making module is in software (gray shadow).

and perform specific computational tasks at sensor endpoints. As for in-sensor computing, individual self-adaptive sensors or multiple connected sensors can directly pre-process sensory information. These innovative solutions effectively minimize redundant data transfers between sensors and external circuits, optimizing raw features in real-time and seamlessly executing vector-matrix multiplication for artificial neural networks.

New materials, devices, algorithms, and architectures need to be further developed for the collaborative design of near-sensor and in-sensor computing. In particular, the rich internal dynamics of emerging volatile memristors can provide a compact and energy-efficient solution to convert sensory information into periodic spiking activities that resemble the encoding mechanism in the biological neural system, which has been successfully demonstrated in the visual and haptic sensory[8–15]. Nevertheless, the dynamics contained in these periodic spiking activities, such as the potential to synchronize through mutual interaction, are not fully exploited to match the complexity of the sensory system. Memristive devices represent a novel component that bridges intrinsic physical dynamics with intricate high-order electronic responses[16], such as oscillation, chaos, and action potential[17–20]. These dynamic devices can interact through electronic coupling, enabling spatial population computing to address classification problems, control problems, and combinatorial optimization problems[21–26]. In these dynamic computing systems, time serves as a distinct physical variable that inherently reveals the system's evolution law, characterized by inherent parallelism. The new paradigm of memristive dynamic computing is poised to bring significant advantages in area and power efficiency, as each device can represent a set of differential equations with state variables rooted in physical processes, thereby reducing the need for a large number of digital circuit operations.

To implement memristive nonlinear oscillators in a dynamic computing system, it's vital to employ resistors/capacitors coupling to explore additional computing resources in synchronous/asynchronous dynamics, facilitating the construction of an oscillatory neural network (ONN). Oscillatory neural network is the embodiment of spatial complexity and can utilize synchronization dynamics to convert information into oscillations' phase, frequency, or waveform. Compared with artificial neural network (ANN), which only uses one-order nonlinear dynamics, ONN captures more dynamics and complexity of the high-order nonlinear dynamic nodes due to its ability to assimilate information in the continuous-time domain. There have been several works applying coupled electric oscillators, deriving dynamic behavior from charge oscillation, spin-torque dynamics, and phase-transition dynamics to undertake recognition tasks, control tasks, as well as solving optimization problems with ultra-low device number and power cost[20,23–25]. However, existing works still face certain limitations. On the one hand, ONN hardware is not yet capable of interacting with the physical environment and collecting sensory information in real-time. On the other hand, output information from the ONN is encoded in the phase patterns, which can be challenging to directly decipher for high-level processing. Thus, a unified framework enabling efficient spatio-temporal pattern processing is yet to be explored.

The proposed solution to the aforementioned challenges draws inspiration from biology. The brain employs an efficient method to handle sensing in the noisy analog domain, demonstrating the ability to discriminate sensory input such as visual movement and sound patterns at extremely fast speed. Taking the visual pathway as an example, a subcortical pathway exists from the front-end retina to the back-end superior colliculus (SC), accounting for the rapid recognition of motion patterns[27] (Fig. 1a). In contrast to the hierarchical feedforward approach of extracting features layer by layer[28], the retinal network in this shortcut adopts a structure combining two dynamic networks[29]. Specifically, a recurrent network is indicated to retain the memory trace of external visual inputs, mapping the spatio-temporal structure of a motion pattern into a specific state of the network. Subsequently, the state of the retina is read out to the second dynamic network, allowing for spatial sampling and information integration[29]. Here, the neurons compete with each other through mutual inhibition until the winner emerges, referring to the category of the input sensory pattern[30].

In this article, we propose and implement a dynamic sensory system based on coupled $VO_2$ oscillators, which adopt near-sensor computing. The comprehensive computing architecture is shown in Fig. 1c, d and will be detailed in the following section. We initially showcase the implementation and characteristics of the $VO_2$ sensory neuron and delve into the synchronization in the ONN realized by the interaction between neurons. The system is composed of two dynamic motifs: an oscillation neural network based on coupled $VO_2$ oscillators for low-level phase difference encoding in hardware, and a decision-making network for high-level processing in software. Finally, we provide experimental demonstrations of touch recognition and a gesture recognition task using the proposed sensory system. This marks the first instance of achieving sensory computing based on coupled memristive oscillators, endowed with the capability to interact with the environment. The $VO_2$ oscillator exhibits high cycle-to-cycle uniformity and it showcases the remarkable capability of reaching oscillation frequencies of up to 2.6 MHz. Three coupled oscillators can encode eight kinds of synchronization modes and have a lower EDP (energy-delay product) of 3.07 pJs. The synchronization mechanism exhibits significant tolerance to device-to-device variation through phase locking, showing the potential of the dynamic computing system.

## Results

### Dynamic sensory oscillation system

The novel dynamic sensory system adopts the theory of synchronized oscillation[31,32]. Memristive oscillators possess the remarkable capability to unify sensing, storage, and computation by dynamically adjusting their intrinsic frequencies in response to sensory signals. These oscillators are further interconnected through weak coupling, forming a synchronization pattern that enables the continuous-time output of pre-processed signals. Within this framework, information is encoded and conveyed through phase differences. As depicted in Fig. 1c, the input sensory stimuli alter the sensing capacitance and determine the intrinsic frequencies of the $VO_2$ oscillators. As a result, the specific sensory pattern is converted to a set of natural frequencies ($f_1, f_2,..., f_n$) that are involved in the dynamic evolution of the network. The oscillatory neurons with the natural frequencies ($f_1, f_2,..., f_n$) interact with each other and evolve towards a collective ground state which refers to the synchronization with stationary phase pattern ($O_1(t), O_2(t),..., O_n(t)$)[33] (Fig. 1c). Here, $s_n$ serves as an abstract representation of the physical stimuli received by neuron $n$ in the ONN, while $f_n$, $O_n(t)$ denotes the natural frequency and voltage output of neuron $n$ in the ONN. Following, the encoded and pre-processed information which is continuous in time from ONN is transmitted to the following decision-making module for further analysis and classification. The decision-making module reads out the phase pattern via a linear weight layer and obtains the classification result through information integration and mutual inhibition.

The decision-making network we employed is a simplified mean-field decision-making model[34,35]. In the decision-making network illustrated in Fig. 1c, there are $n$ neurons, each representing one of the classification decisions of the sensory pattern. Among the neurons, mutual inhibition is introduced to make them compete with each other and determine the sole winner as the final result. The model can be

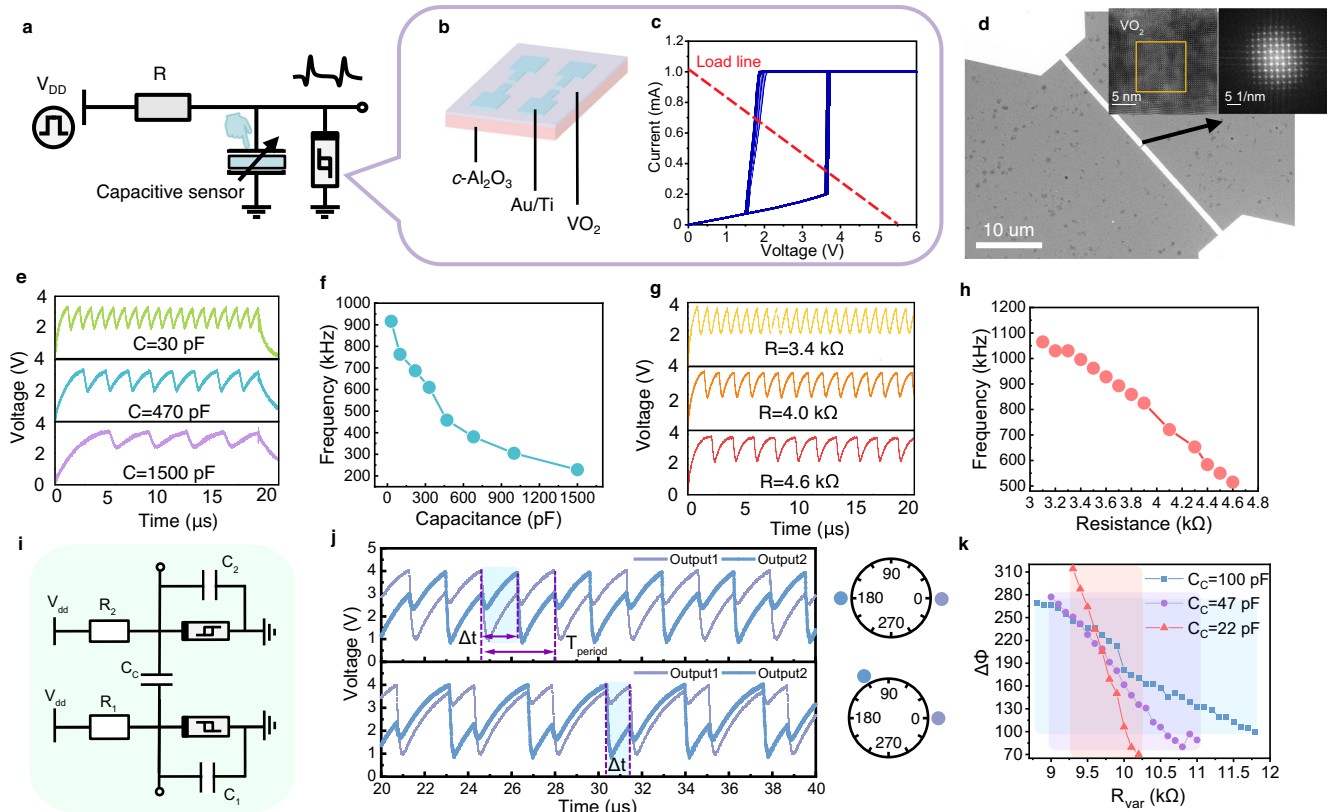

**Fig. 2 | The dynamic of oscillatory neuron. a** Schematic of the proposed sensory neuron based on VO₂ memristor second-order oscillators. **b** Schematic of the planar VO₂ device which is epitaxial growth on c-Al₂O₃. **c** Electric characteristic under bidirectional voltage sweeping. The VO₂ device performs hysteresis switching with two thresholds ($V_{th}$ and $V_{hold}$). The oscillation load line is shown in red. **d** The SEM and TEM characterization of the VO₂ devices with length of 200 nm. **e** Typical continuous oscillation output of the sensory neuron with different parallel capacitances. **f** The relationship between the parallel capacitance and the output frequency. **g** Typical continuous oscillation output of the sensory neuron with different series resistances. **h** The relationship between the series resistance and the output frequency. **i** Circuit diagram (in green shadow) of two capacitive coupled oscillatory neurons. **j** Typical experimental results of the synchronization patterns and phase differences. The oscillatory neurons with identical natural frequencies tend to oscillate in an out-of-phase manner in the upper panel. The blue shadow means the spiking time difference △t. **k** The relationship between △φ and series resistor as a variable $R_{var}$. Here $R_{var}$ determines the frequency mismatch between the reference neuron with fixed natural frequency and that with variable frequency. The red, purple, and blue shadow represent the different coupling ranges when $C_c$ = 22 pF, 47 pF, 100 pF.

mathematically described by the following equations[35]:

$$x_i(t) = J_E s_i + \sum_{j \neq i}^{N_{dm}} J_M s_j + I_i \tag{1}$$

$$r_i = \frac{\beta}{\gamma} \ln\left[1 + e^{\frac{x_i - \theta}{\alpha}}\right] \tag{2}$$

$$\tau_s \frac{ds_i}{dt} = -s_i + \gamma(1 - s_i) r_i \tag{3}$$

where $x_i$ and $r_i$ denotes the synaptic input and neuronal activity of the $i_{th}$ neuron. $s_i$ represents the synaptic. The Eq. (1) describes the neurons in the decision-making module that receive synaptic current from themselves feedback $J_E s_i$, other neuron $J_M s_j$ and the input from the former layer $Z_i$. The Eq. (2) describes the neuron's nonlinear activation function with a threshold θ. The Eq. (3) describes the slow dynamics of the synaptic current. Specifically, the synaptic input $x_i$ described by Eq. 1 is composed of three parts, which are the self-excitation $J_E s_i$ ($J_E \geq 0$), total recurrent input from other neurons with $J_M \leq 0$ indicating mutual inhibition and the feedforward input $I_i$ from the ONN module. The neuronal activity $r_i$ can be further calculated by $x_i$ using the nonlinear activation function in Eq. 2 with parameters α, β, γ, and

threshold θ. The slow dynamics of the synaptic current originating from the activity-dependent receptors is formulated in Eq. 3, which plays a crucial role in the spatiotemporal information processing of the decision-making network. The time constant $\tau_s$ ($\gg 1$) in Eq. 3 controls the time window for integrating input over time by the decision-making neurons. The first network's output is the voltage oscillation signals, which is the (($O_1(t)$, $O_2(t)$, ..., $O_n(t)$) in Fig. 1c, d. First, the voltage oscillation signals through linear combination. The impact of the linear combination is to adjust the signal to an appropriate range. The weight of the linear combination can be trained by Force learning[35]. Then the weighted signals are the input to the decision-making module as shown in Eq. 1. The expected output **r** of the decision-making module is that one neuron fires with maximum strength while the others fire less.

The specific hardware and software mapping of the dynamic sensory system is shown in Fig. 1d. A VO₂ threshold device is employed to construct an oscillator circuit, where the frequency can respond to changes in the sensing capacitor. Then, multiple VO₂ oscillators with sensing functions can be capacitively coupled. The oscillator will synchronize to the same frequency under suitable conditions. Meanwhile, the information of output voltage oscillation transitions from frequency to phase difference after coupling. This design utilizes the interaction of dynamic devices in space and encodes the sensory information in the phase pattern after electrical coupling. The coupled

oscillation voltage signals can then be transmitted to the decision-making model for further processing.

## Dynamics of VO₂ sensory oscillator

The Mott material $VO_2$ features a fast and high-uniformity insulation-metal phase transition dynamic[15], encompassing a temperature-sensitive electronic transition and a structural transformation from a low-temperature, low-symmetry monoclinic (M1) phase to a high-temperature, high-symmetry tetragonal (R) phase[36]. This physical phase transition dynamic is promising for constructing neuromorphic applications. Compared with $NbO_2$, $VO_2$ has a smaller volumetric enthalpy, indicating lower power consumption due to its relatively low transition temperature (about 340 K)[37]. The artificial neuron based on $VO_2$ has been demonstrated to generate multiple spiking modes, showing the potential for computing at the edge of chaos[38]. Here, we construct a voltage-driven artificial sensory neuron as a primary computing cell with a monocrystalline $VO_2$ memristor, a constant resistance and a sensing capacitor, as illustrated in Fig. 2a. The core function is to detect external environmental stimuli via capacitive receptors and convert the information into electric spikes, as depicted in Fig. 2a. The $VO_2$ thin film is grown on c-$Al_2O_3$ substrates in an epitaxial manner by pulsed-laser deposition (PLD) technique. The $VO_2$ memristive device adopts a planar configuration with the switching layer positioned between the Au electrodes as shown in Fig. 2b. The Scanning Electron Microscope (SEM) and Cross-sectional Scanning Transmission Electron Microscope (STEM) images in (Fig. 2d and Supplementary Fig. 1) confirm the structure of the fabricated device.

The $VO_2$ memristor exhibits a volatile nonlinear dynamic, whose internal state variable can be defined as the device's temperature[39]. When the $VO_2$ device is applied with a quasi-static voltage sweep, the corresponding current exhibits a hysteresis switching response (Fig. 2c). Initially, the device stays in a semiconducting monoclinic state, a high resistive state (HRS, $R_{off}$). When the voltage across exceeds a threshold value, denoted as $V_{th}$, the metal-insulator transition (MIT) takes place and $VO_2$ turns to a metallic rutile phase, a low resistive state (LRS, $R_{on}$). An abrupt increase in the current can be observed which also exhibits a saturation in our case due to the compliance during measurement. Given the orders of magnitude change in resistance (from 20 kΩ to 300 Ω), the compliance current is applied to safeguard the devices. The quasi-static voltage sweep without compliance is shown in Supplementary Fig. 2. During the voltage retrace process, when the voltage descends below the critical value for maintaining the metallic state, denoted as $V_{hold}$, the $VO_2$ layer undergoes a reverse process and returns to its high resistive state which corresponds to the sharp decrease in current. The $VO_2$ device is a bidirectional symmetric planar device, demonstrating a symmetrical response to positive and negative voltage scans (as shown in Supplementary Fig. 3). Its conductance rapidly decreases after removing the applied voltage (automatically from on-state to off-state). Therefore, it is a volatile non-polar memristive device. The underlying physics in MIT of the $VO_2$ device has attracted wide interest yet[40,41]. The switching of $VO_2$ is believed to involve complex electronic and structural phase transitions[42].

The monocrystalline $VO_2$ memristor provides high cycle-to-cycle uniformity (Supplementary Fig. 4). The deviation of the threshold voltage is 0.00021 and the deviation of the hold voltage is 0.00013. The high cycle-to-cycle uniformity is crucial for the practical oscillation demonstration because the oscillator must consistently output a constant frequency. 40 $VO_2$ devices are characterized to study their device-to-device variation, which will affect different natural frequencies of $VO_2$ oscillators, as shown in Supplementary Fig. 5. The threshold voltage under quasi-static voltage sweep varies from 3.55 V to 4.10 V and the hold voltage varies from 1.52 V to 1.96 V. Devices with similar performance are more likely to establish out-of-phase synchronization after capacitive coupling. Furthermore, the $VO_2$ device's electrical response to pulse stimuli is important to dynamic oscillation.

Supplementary Fig. 6a shows the testing circuit diagram of the devices' switching time. We gradually increase the pulse voltage from the threshold voltage and recode the time difference between voltage input and current response. The testing result is shown in Supplementary Fig. 6b–h. It can be observed when the voltage increases from the threshold voltage of the device (4.5 V), the device will have a transition from off-state to on-state. The switching time will first have a rapid reduction from 300 ns to 120 ns when the pulse voltage is around the $V_{th}$. The similar effects can also be observed in other threshold devices, like Ag-based atomic-switching threshold devices and B-Te-based Ovonic threshold devices[43]. When the applied voltage is greater than 4.8 V, the switch speed saturates (around 115 ns). When the voltage pulse is removed, the $VO_2$ device will quickly switch back from on-state to off-state. Voltage pulses with different voltage amplitudes are applied on the $VO_2$ device for 10 μs and the time difference between voltage input and current response is recorded, as shown in Supplementary Fig. 7b–h. It can be observed that the switching time is settled around 90 ns. The amplitude of the applied voltage has less influence on the device's retention process. We plot the switching time of off-state to on-state and on-state to off-state in Supplementary Fig. 8. The devices' switching time and retention properties limit the overall working frequency. As for the $VO_2$ device in this work, the maximum working frequency is around 2.6 MHz. Leveraging the pronounced nonlinear transition characteristics of the $VO_2$ device, vital behaviors of biological neurons can be emulated and more nonlinear coupling behaviors can be constructed further.

The $VO_2$ device is connected with resistance and sensing capacitance to form an oscillation circuit, as shown in Fig. 2a. When subjected to a constant voltage input $V_{dd}$, the $VO_2$ device is expected to cyclically transition between its HRS and LRS, generating an oscillating signal at the output node. The dynamics of the $VO_2$ oscillator circuit can be described as the following ordinary differential equation:

$$C_s \frac{\mathrm{d}V_m}{\mathrm{d}t} = \frac{V_{dd} - V_m}{R_L} - \frac{V_m}{R_{VO_2}} \tag{4}$$

Where $C_s$ is the sensing capacitance connected in parallel to the $VO_2$ device. $V_m$ is the output voltage across the $VO_2$ memristive device. Specifically, the initial high resistance of the $VO_2$ results in the accumulation of charge in the capacitance until the device's voltage arrives $V_{th}$, allowing the transition towards the metal phase to take place. Subsequently, due to the discharge of capacitors, when the devices' voltage is lower than $V_{hold}$, the low resistance state can no longer sustain sufficient voltage to hold the metal state, causing the device to revert to its HRS. It should be pointed out that the constraint of the series resistance is implied above to ensure an appropriate working voltage range for the $VO_2$ device. It is necessary to ensure that the device's voltage higher than $V_{th}$ at its HRS and lower than $V_{hold}$ at its LRS. In the $VO_2$ thin film, the switching voltage variation arises from the generation and dissipation of Joule heating and can be effectively minimized to maintain a low level[42].

The output frequency dependence on circuit parameters is also investigated through electrical measurements. As shown in Fig. 2e, f, the frequency of the output spike is influenced by the value of parallel capacitor. As we increase the capacitance, the oscillating frequency is reduced accordingly ascribed to a longer charging time to reach the threshold. Output waveforms corresponding to different parallel capacitance values can be found in Fig. 2e. According to results in Fig. 2g, h, the value of the series resistor is also negatively related to output frequency which lies in the fact that a larger resistance diminishes the charging current. Outputs corresponding to different series resistances are shown in Fig. 2g.

The artificial haptic neuron equipped with a capacitive sensor can efficiently respond to external stimuli. Supplementary Fig. 9a shows the characteristic of the commercial pressure sensor where the pressing

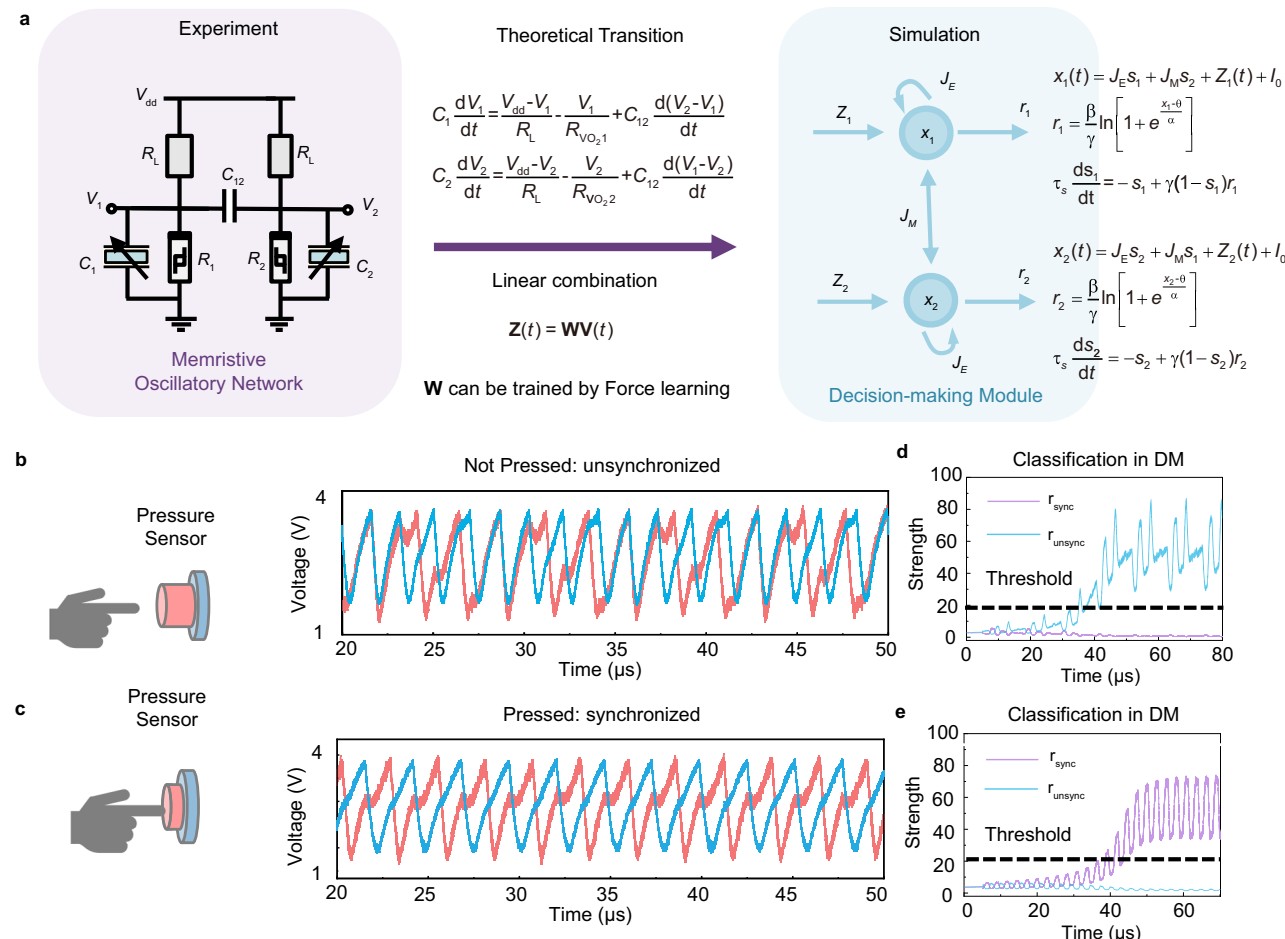

**Fig. 3 | Touch recognition with the bio-inspired sensory recognition system. a** A computing flow chart of the coupled VO₂ memristive oscillators, the decision-making module, and the mathematical link between them. The purple shadow means in experiment and the blue shadow means in simulation. **b** The phase pattern without touch event. The outputs of the two neurons in ONN are not synchronized to the identical frequency. **c** The phase pattern with touch event. The outputs of the two neurons in ONN are locked in frequency with stable difference in phase. **d** Outputs of the decision-making network corresponding to the non-synchronized condition (without touch). **e** Outputs of the decision-making network corresponding to the synchronization pattern (with touch).

action causes an increment in its capacitance. The haptic neuron can thereby convert the pressure information into the oscillating frequencies. The resultant output in time domain is shown in Supplementary Fig. 9b while the single-sided amplitude spectrum through Fast Fourier Transform (FFT) is depicted in Supplementary Fig. 9c, jointly validating the function of our electric haptic neuron. In addition, we construct another type of haptic neuron to sense the stretching level. The capacitance of the adopted sensor is directly proportional to the degree of stretch (Supplementary Fig. 10a), further accounting for the decreasing of frequencies (Supplementary Fig. 10b).

The sensory neurons in our ONN exchange and integrate information through electrical coupling, as illustrated in Fig. 2i; where the output nodes of the pair of neurons are connected via a capacitor. The processing results can be extracted from the phase pattern after synchronization. Specifically, synchronization is a dynamic state emerging from the ONN system, which can be described as the adjustment of rhythms among different oscillatory neurons due to their weak interactions[44,45]. Generally, the synchronized neurons will hold an identical frequency with a stationary phase pattern. Theoretically, there are some requirements for the synchronization, including (1) the oscillating elements involved are nonlinear self-sustained oscillators, which have stable limit cycles in phase space and therefore lead to immunity to small perturbations; (2) the interaction between these oscillatory neurons should be weak enough in case of breaking their oscillating individuality. In capacitive coupling configuration, the two

oscillatory neurons with identical natural frequencies tend to lock their phase with each other to an out-of-phase pattern as observed in the upper panel of Fig. 2j, where the phase difference is calculated by measuring Δt between the peaks of the outputs from the two channels. The reason of the out-of-phase result lies in the fact that the two electrodes of the coupling capacitors always attract charges of opposite polarity which results in a mutual inhibitory effect between their oscillatory behavior. Moreover, the neurons modify their phase pattern depending on the difference between their natural frequencies and the strength of the mutual coupling. Specifically, as shown in the lower panel of Fig. 2k, if we decrease the natural frequency of the second neuron (whose output is denoted as output₂ in the blue line) by increasing its series resistance, the two neurons will synchronize with a lower frequency and a smaller phase difference. Supplementary Fig. 4 shows the phase relationship of two oscillatory neurons coupled by different parallel capacitors in synchronization.

Generally, Synchronization can only be achieved when the initial frequency difference between the two oscillatory neurons is within a certain range. A large capacitor can provide stronger coupling strength to synchronize neurons that are more inconsistently paced (Fig. 2k). It's worth noting that the synchronization between the artificial neurons can also be realized with resistive coupling, which will lead to in-phase coupling. It should be pointed out that the natural frequencies ($f_1, f_2, ..., f_n$) are intrinsic properties of the oscillatory neurons that modulate the final phase pattern after the interaction. However, they

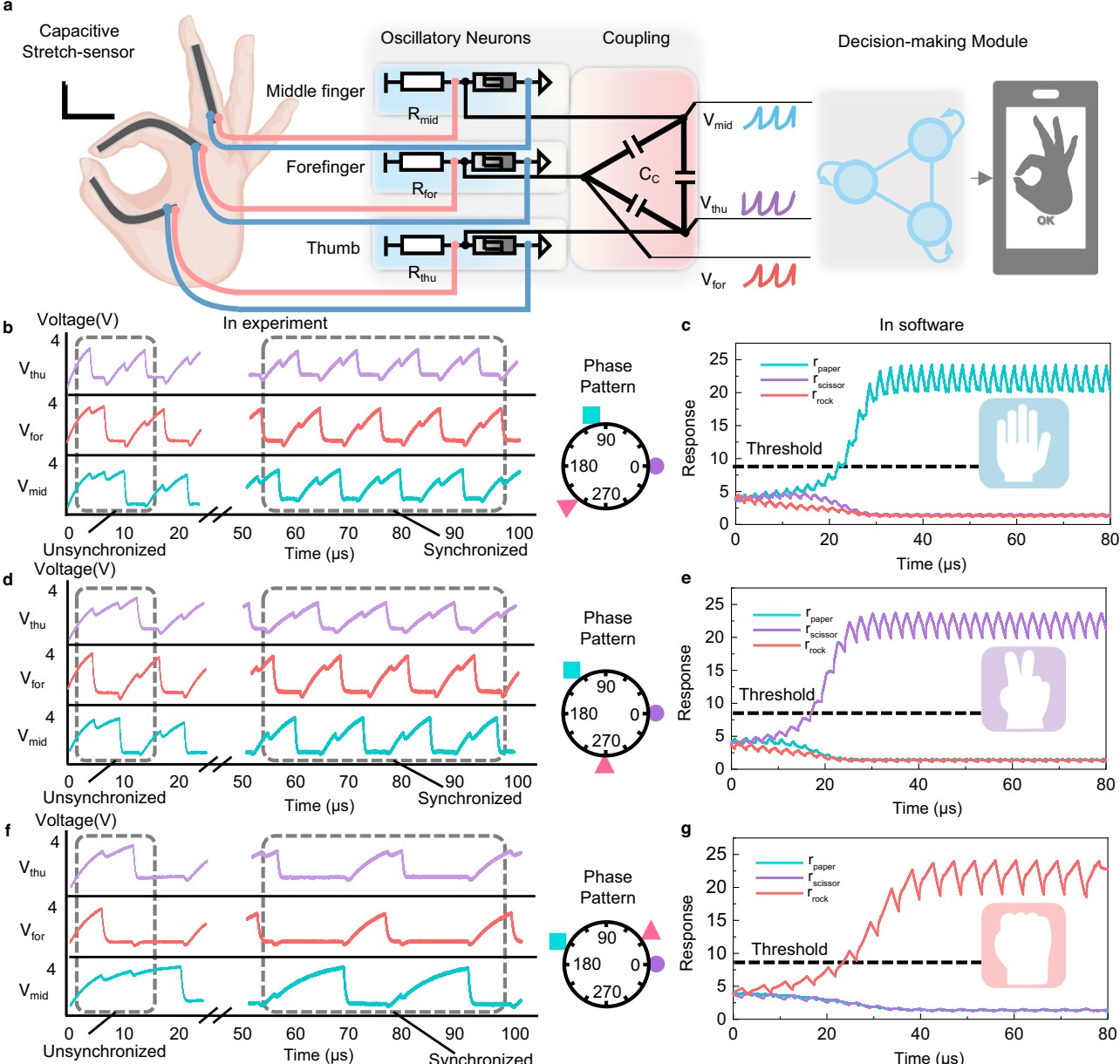

**Fig. 4 | Gesture recognition with the bio-inspired sensory recognition system.**
**a** An illustration of the sensory system for gesture recognition. The capacitive-stretch sensors receive the change of length and are parallel connected to the VO$_2$ oscillators, which are coupled by coupling capacitances. Then the oscillation signals are input to the decision-making module in the software and output the classification results. **b** Typical phase pattern from the ONN with an input gesture of "paper" (from unsynchronized to synchronized). **c** Outputs of the decision-making network corresponding to "paper". **d** Typical phase pattern from the ONN with an input gesture of "scissor" (from unsynchronized to synchronized). **e** Outputs of the decision-making network corresponding to "scissor". **f** Typical phase pattern from the ONN with an input gesture of "rock" (from unsynchronized to synchronized). **g** Outputs of the decision-making network corresponding to "rock".

are invisible during the information processing and can be only measured when the neurons are individual without coupling in the network. The extracted features are contained in the phase rather than frequency of output ($O_1(t)$, $O_2(t)$,..., $O_n(t)$). Since the ONN generally needs only a few periods to be synchronized and the frequencies are not required to readout, the whole encoding and nonlinear feature extracting process seems to happen simultaneously. The total processing time of ONN equals the time to achieve synchronization, showcasing its high time efficiency.

### Experimental implementation of touch recognition

Environmental variables can interact within memristive devices and realize high-order complex behaviors (Supplementary Note. 1). When

memristive devices are coupled, the dynamics can be further utilized for coding and computation. Under different sensing capacitances, the coupling system can represent various dynamic behaviors such as synchronization, frequency, phase, and even waveform. This feature demonstrates the dynamic high-order memristive circuit's ability to interact with the environment and manifest unique behaviors under different environment stimuli.

We first demonstrate the function of our sensory recognition system with a simple binary classification experiment where the system needs to distinguish whether a touch event occurs. A flow chart of the coupled VO$_2$ memristive oscillators (in experiment), the decision-making module (in software), and the mathematical link between them is shown in Fig. 3a. We can have the following ordinary differential

equations to describe the coupled oscillators:

$$C_1 \frac{dV_1}{dt} = \frac{V_{dd} - V_1}{R_L} - \frac{V_1}{R_{VO_2 1}} + C_{12} \frac{d(V_2 - V_1)}{dt} \quad (5)$$

$$C_2 \frac{dV_2}{dt} = \frac{V_{dd} - V_2}{R_L} - \frac{V_2}{R_{VO_2 2}} + C_{12} \frac{d(V_1 - V_2)}{dt} \quad (6)$$

These equations imply the synchronization under proper parameters. With appropriate coupling capacitances, the oscillators with different intrinsic oscillation frequencies can oscillate with the same frequency but in different phases. When we change the sensing capacitors, the system's phase different mode will also change, which encodes the sensing information. Besides, a detailed software algorithm flow about the decision-making module in post-processing is shown in Supplementary Fig. 15.

In this demonstration, we set up two oscillatory neurons in the ONN network to encode and process the haptic information. Specifically, one of the neurons is served as a reference neuron with a fixed natural frequency, that is the parallel capacitor in this neuron is configured as a fixed one. Whereas the other neuron is a sensory neuron equipped with a pressure sensor whose characteristic is shown in Supplementary Fig. 9. If no touch event happens, the capacitive sensor maintains its original state with a relatively small capacitance. In this condition, the sensory neuron exhibits a relatively high frequency accounting for a large mismatch with the reference neuron. Consequently, these two neurons cannot be synchronized to an identical frequency through interaction and adjustment as shown in Fig. 3b. On the contrary, when stress is applied to the pressure sensor, the compressed distance between the electrodes leads to an increment in the sensor capacitance. The oscillation of the sensory neuron is thereby slowed down which allows the network to synchronize through the evolution. The synchronization pattern corresponding to a typical trial of touch event is depicted in Fig. 3c.

There are two competing neurons in the subsequent decision-making module with each one representing one preference. These two decision-making neurons are denoted as $n_{syn}$ and $n_{unsyn}$ with response $r_{syn}$ and $r_{unsyn}$ respectively. The decision-making network is trained to distinguish the synchronization pattern and the disorder pattern. When the neuron $n_{syn}$ is activated, it means a synchronization pattern is detected by the decision-making module which indicates a touch event and vice versa. The detailed information about training can be found in the method section. Figure 3e shows the evolution of the decision-making network when the synchronized pattern in Fig. 3c is received. At the beginning, the activities of both neurons are low and intermingled with each other. As time goes on, owing to integration of input information and competition enabled by mutual inhibition, the neuron $n_{syn}$ eventually wins which points to a touch event. The opposite case is also depicted in Fig. 3d. The detailed processes and formulas of this high-level processing can be found in Supplementary Fig. 15.

**Experimental implementation of gesture recognition**
When expanding the oscillators from two to three, Supplementary Fig. 14 shows the synchronization dynamic between three oscillatory neurons through capacitances under different oscillators natural frequencies. We keep the two oscillators' natural frequencies the same and change the other oscillator's natural frequency by changing the series resistor $R_3$. The $R_1 = R_2 = 10.2\,k\Omega$. When changing $R_3$, the phase differences present monotonic changes when $R_3 < 10.8\,k\Omega$. Especially, when $R_3 = R_1 = R_2 = 10.2\,k\Omega$, the three coupled oscillation frequencies exhibit a phase difference of $\pi/3$ and $2\pi/3$. It shows the $VO_2$ device-to-device variation can be suppressed by the coupling capacitors.

We further demonstrate a gesture recognition task, the "rock, paper, scissors", using our bio-inspired recognition system. For this demonstration, we use three sensory neurons in the ONN to receive the stretching information from the thumb, forefinger, and middle finger, as illustrated in Fig. 4a.

Then we can have the following ordinary differential equations:

$$C_{s1} \frac{dV_{thu}}{dt} = \frac{V_{dd} - V_{thu}}{R_{thu}} - \frac{V_{thu}}{R_{VO_2 1}} + C_{p12} \frac{d(V_{for} - V_{thu})}{dt} + C_{p13} \frac{d(V_{mid} - V_{thu})}{dt} \quad (7)$$

$$C_{s2} \frac{dV_{for}}{dt} = \frac{V_{dd} - V_{for}}{R_{for}} - \frac{V_{for}}{R_{VO_2 2}} + C_{p23} \frac{d(V_{thu} - V_{for})}{dt} + C_{p12} \frac{d(V_{mid} - V_{for})}{dt} \quad (8)$$

$$C_{s3} \frac{dV_{mid}}{dt} = \frac{V_{dd} - V_{mid}}{R_{mid}} - \frac{V_{mid}}{R_{VO_2 3}} + C_{p13} \frac{d(V_{thu} - V_{mid})}{dt} + C_{p23} \frac{d(V_{for} - V_{mid})}{dt} \quad (9)$$

These three neurons are denoted as $n_{thu}$, $n_{for}$ and $n_{mid}$ with their output $V_{thu}$, $V_{for}$ and $V_{mid}$, accordingly. A visual representation of the experiment setup, along with block diagrams of the components used is shown in Supplementary Figs. 11, 12. The sensory neurons are powered by the (10 V, 300 μs) long voltage pulse as $V_{dd}$, and the series resistors are configured as $R_{thu} = R_{for} = R_{mid} = 8.2\,k\Omega$. The coupling capacitors of 100 pF are adopted. We power up the memristive ONN as soon as the gesture is ready. The random initialization will give a random phase output. To ensure that the initial state doesn't affect the phase difference after coupling, we set a start-up circuit to make sure the phase modes of each output node remain stable in multiple experiments. The set-up circuit is a simple voltage divider circuit to determine the initial stage of the coupled oscillators. The total circuit diagram of ONN with a start-up circuit is shown in Supplementary Fig. 16. We supplement the transient process at the beginning of coupling in Supplementary Fig. 17 to show initial signals and their synchronization. The transient process before the stable synchronization is around 20 μs.

The stretching sensor covers the outer surface of the finger so that the bending of the finger is encoded into the natural frequency of the sensory neuron according to Supplementary Fig. 10. As the ONN evolves towards the stationary state, the input gesture will be mapped to the corresponding synchronization pattern. The phase patterns responding to the typical trials of "paper", "scissor" and "rock" are shown in Fig. 4b, d, f, respectively. In these examples, the output phase patterns are ($\phi_1 = 0°$, $\phi_2 = 236°$, $\phi_3 = 105°$), ($\phi_1 = 0°$, $\phi_2 = 270°$, $\phi_3 = 126°$) and ($\phi_1 = 0°$, $\phi_2 = 40°$, $\phi_3 = 165°$). It could also be found that the resultant oscillatory pattern of "rock" has a lower synchronized frequency of 45 kHz, which lies in the fact that the natural frequencies of all the sensory neurons are low due to the obvious bending of these three fingers. Phase patterns from ONN in case of more gesture input can be found in Supplementary Figs. 19 and 20. Supplementary Table 1 lists the phase mode from experimental ONN to three dynamic sensory neurons. For mathematical verification of the coupling among sensory nodes, the FFT is applied to analyze the frequency consistency, as shown in Supplementary Fig. 18. The FFT result shows the consistency of frequency in all eight cases that 3 coupled oscillators can encode.

The readout matrix of the decision-making network is also trained by FORCE learning. Here, three competing neurons which are denoted as $n_{paper}$, $n_{scissor}$ and $n_{rock}$ are employed to represent the different classification decisions. As shown in Fig. 4c, e, since the phase patterns originated from the "paper" and "scissor" are close to each other, their

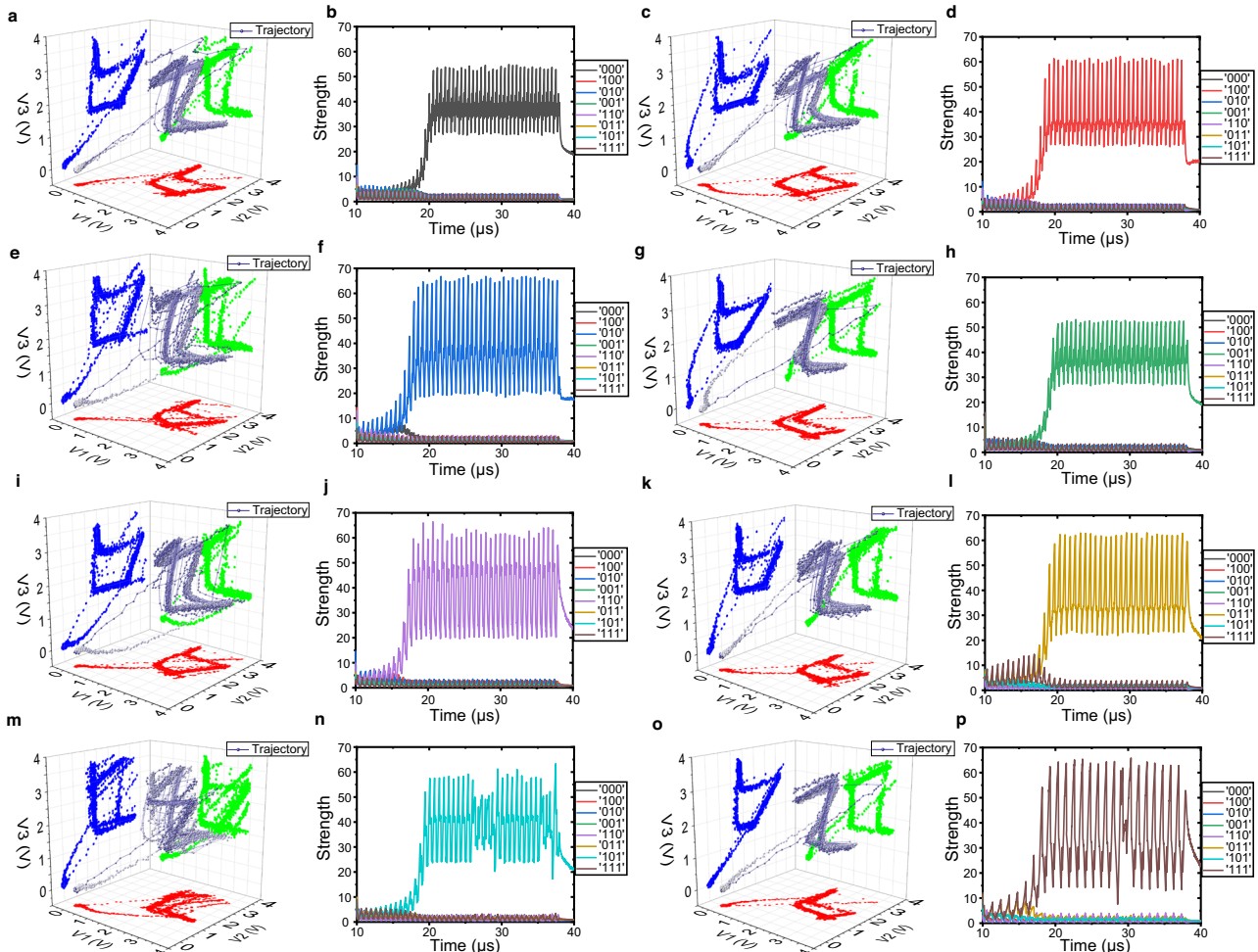

**Fig. 5 | The trajectories in the phase plane of three coupled VO$_2$ oscillators in eight coupled oscillation modes and the decision-making classification results.** **a** The experimental output voltage trajectory in the phase plane of case '000'. The red, green and blue dots mean the trajectory projection on x-y, x-z, y-z plane. **b** The decision-making classification results in the software of case '000'. The different colors represent the different responses to preferred cases. Besides, the experimental trajectories in the phase plane of case '100', '010', '001', '110', '011', '101', and '111' correspond to (**c, e, g, i, k, m, o**). The decision-making classification results in the software of case '100', '010', '001', '110', '011', '101', and '111' correspond to (**d, f, h, j, l, n, p**). Three coupled VO$_2$ dynamic devices can encode eight sensory modes in phase space to form limit cycles, which can be classified by the decision-making module.

corresponding neurons are initially intermingled together. After integrating the evidence for several periods, the neuron that represents the correct decision eventually prevails over the others and exceeds the threshold. And the activities of the other neurons are suppressed to a very low value through mutual inhibition. The case of "rock" could also be found in Fig. 4g. The successful demonstrations of touch recognition and gesture recognition experimentally have proven the great potential of our bio-inspired system for accomplishing the sensory recognition task in general. The delay time before stable coupled is about 20 μs. The system has a low energy-delay-product of 3.07 pJs, which is reduced by 1000 × compared to classic CMOS methods[46].

Couple VO$_2$ oscillators can have stronger computing superiority compared to the other methods including non-volatile memristors and CMOS circuits[15,46–49]. A detailed comparison table with previous works is shown in Supplementary Table 2. The volatile memristive devices including VO$_2$ and NbO$_2$ suffer from the large device-to-device variation. Multiple VO$_2$ devices exhibit certain device-to-device variations even at the same size, which can be disadvantageous when used as individual oscillators. However, the coupling oscillation strategy we adopt enables the synchronization of two oscillators with distinct intrinsic oscillation frequencies to a common frequency. Subsequently, we encode and classify them based on their phase patterns.

The alteration in sensing capacitance exerts a more pronounced impact on the phase difference after the oscillators are coupled. Consequently, this strategy serves to mitigate the impact of device-to-device variation. Besides, due to the high threshold voltage and low on-state resistor, the VO$_2$ oscillators have a high power, which can be improved in the future. The delay time in coupled oscillation can be defined as the time of the transient process from the initial state to the stable coupling state, which is only 20 μs in this work (as shown in Supplementary Fig. 3). Most importantly when expanding to larger systems, $n$ coupled VO$_2$ oscillators can encode $2^n$ distinguishable modes while the other dynamic processing methods may require exponential device consumption. In this work, we experimentally demonstrate the three coupled VO$_2$ oscillators, and the eight encoding modes are shown in Fig. 5. Due to coupling characteristics, the output voltage signals can form clear trajectories in phase space, which reflect the classifiable modes as shown in Fig. 5a, c, e, g, i, k, m, o. The trajectories and their projections on x-y, y-z, and x-z planes present the dynamic of stable limit cycles, whose shapes reflect the phase differences of different oscillators. The eight kinds of limit cycles attach great importance to dynamic computing based on memristors. It shows the actual physical memristors with variations can be accurately controlled in population dynamics. To prove that the eight feature

modes can be classified in the proposed dynamic systems, we expand the scale of the decision-making module to 8 neurons, and each neuron presents one mode. The software simulation is shown in Fig. 5b, d, f, h, j, l, n, p. When voltage signals in one mode are linear weighted and input into the module, the neuron corresponding to this combination will fire with maximum and suppress the other neurons. The results show the decision-making module can handle such dynamic classification problems while other methods like recurrent neural networks need hundreds of neurons in hidden layers.

## Discussion

The brain is a highly complex nonlinear system. The artificial neural network on hardware, which applies linear vector-matrix multiplication and nonlinear activation function, suffers from high power consumption and limited intelligence compared to the biological counterpart. This defect occurs due to the oversimplification and abstraction of the neural system, leading to the loss of many key dynamics in the brain. Generally, the order of neurons and the way they interact determine the complexity and dynamics of the system. Emerging dynamic memristive devices can capture neural behaviors and further reduce area and power consumption. Thus, ONN used in this work replaces hierarchical structure and a huge number of parameters with complex dynamics brought by recurrent connections of oscillatory neurons. Although sensory neurons in our system are designed to capture mechanosensory information, the system can be easily extended to other sensory inputs or multi-sensory inputs by replacing the capacitive sensor (or select the appropriate resistive sensor based on the input signal and connect it in series with the device). It should be pointed out that the recurrent network is required to map different spatiotemporal patterns into spatially separated neural states, can also be replaced by reservoir computing. Recent research has also shown the potential of nonlinear volatile memristors to implement reservoir computation in a compact and efficient way[48,50,51]. With regard to the decision-making module, this part is currently realized in software. The key to building a hardware decision-making network is to emulate the inhibition function with electric devices which still needs to be explored.

Regarding nonlinear $VO_2$ memristive oscillators, our primary focus is on leveraging their coupled oscillation characteristics to investigate the impact of expanding spatial complexity on computing capacity. The dynamic process of metal-insulator transition of $VO_2$ can also be affected by other environment signals such as irradiation and environment temperature. The transition can be driven by near-IR excitation and has a faster transition speed of about 75 fs[52,53]. Besides, this currently the transition temperature (67 °C) is close to room temperature, which may cause variations of oscillation frequency at different times. It can be optimized by increasing the phase transition temperature through doping $Ge^{4+}$ to change phase stability[54,55]. The $VO_2$ memristive device's specific compositional, structural, and mesoscale levers are supposed to be explored for tuning phase stabilities, transformation pathways and nucleation mechanisms[36]. In the current iteration of our work, the sensing devices are connected to $VO_2$ devices using breadboards and cables, as depicted in Supplementary Figs. 8, 9. There is potential benefit for integrating $VO_2$ devices and sensing devices on a single substrate. The pressure sensor array can be made of MXene on flexible substrates[56]. MXene is a 2-dimensional metal carbide/nitride exhibiting conductivity changes in response to external pressure. Besides, $VO_2$ devices can also have certain sensing functions. For example, environment temperature will change the threshold voltage of $VO_2$ devices, affecting the intrinsic oscillation frequency and further affecting the phase difference after coupling. Thus, $VO_2$ devices inherently possess the capability to undertake certain sensing functions, such as detecting changes in light, stress, and temperature. Furthermore, the Mott device ($VO_2$ and $NbO_2$) can be fabricated at relatively low temperatures (< 300 °C), making them compatible with 3D integration techniques.

In summary, we propose and implement a bio-inspired sensory recognition system combined an oscillatory neural network (ONN) based on $VO_2$ oscillatory neurons in experiments for pre-processing and a decision-making network in simulation for post-processing. The ONN integrates sensing, storage and computation and can control its various dynamic behaviors by changing the intrinsic frequency of the oscillator through sensing signals. The synchronization is generated through weak coupling. The dynamic sensory system based on coupling memristive oscillators is achieved for the first time, with the ability to interact with the environment, demonstrating the potential of nonlinear dynamic system computing. It is the first time to introduce decision networks to process the continuous time signal output by the memristor oscillators, which can use a biologically plausible approach to identifying the information from high-order memristive oscillations. We also apply Force learning as a better training method, which is more suitable for automatically processing classification tasks under continuous spatiotemporal signals. Compared with traditional hardware, the dynamic sensory system has a significant improvement in power consumption evaluation according to the experimental sensory recognition tasks. This work serves as a pioneering step toward the development of a more efficient neuromorphic sensory system based on nano-scale nonlinear dynamics.

## Methods

### Device fabrication

The 20 nm $VO_2$ films were grown on c-$Al_2O_3$ substrates in an epitaxial manner by pulsed-laser deposition (PLD) technique using a 308 nm XeCl excimer laser operated at an energy density of about 1 J/cm² and a repetition rate of 3 Hz. The $VO_2$ films were deposited at 530 °C in a flowing oxygen atmosphere at the oxygen pressure of 2.0 Pa. Then, the films were cooled down to room temperature at the speed of 20 °C/min. The deposition rate of $VO_2$ thin films was calibrated by X-ray Reflection (XRR). The electrodes, which are composed of Au (40 nm) and Ti (5 nm) with a distance of 400 nm, were patterned with electron beam lithography (EBL) along with electron beam evaporation and lift-off. More than two hundred devices were fabricated.

### Electrical measurement

The electrical measurements, including the DC test of the $VO_2$ memristor and the pulse test of the sensory neurons, are conducted using an Agilent B1500A semiconductor parameter analyzer. The Agilent B1500A is also employed to provide the long voltage pulse during the experiments of touch recognition and gesture recognition. All the oscillatory waveforms are captured by the RIGOL MSO8104 digital storage oscilloscope.

### Training procedure

A decision-making (DM) network is applied to classify temporal patterns from coupled $VO_2$ oscillations in simulation[35]. The neurons in this network are self-excited and mutually inhibited, resulting in a winner-take-all characteristic. The neurons accumulate evidence from inputs over time and the one with a stronger input will eventually exhibit a heightened response which corresponds to the inferred choice. The number of neurons depends on the task scale. For synchronized/unsynchronized classification, a similar network with only 2 input neurons and 2 DM neurons was used. For phase pattern classification, the network consisted of 8 input neurons corresponding to the 8 coupled oscillators. These input neurons were fully connected to 8 DM neurons which represent the 8 categories. The network parameters are listed in Supplementary Table 3. The detailed post-processing flow with formulas and explanations is shown in Supplementary Fig. 15. In the inference process, the voltage oscillation signals from the coupled oscillators first pass through the linear combination layer and then input into the decision-making module. In the training process, we set the training target output and compute the error function between the

target and the actual output. Then the error function to adjust the weight of the linear combination with an update matrix. For each input pattern, a few periods of oscillation somewhere towards the end of the measured data were taken as training samples. These samples were then concatenated and repeated in the time dimension to form a complete training input. As for the test inputs, data starting from random initial points somewhere near the beginning were taken. This ensures that the network uses mainly unseen data during testing. Using BrainPy, we trained the fully connected weights by applying FORCE learning[57,58]. The goal was to provide the correct DM neuron with a larger linear combination of inputs as compared to the other DM neurons. This was done by setting a positive constant as the target function to be received by the correct DM neuron and a negative constant as the target function to be received by the other DM neurons. In these tasks, the respective networks correctly classified the input patterns.

## Data availability

All data supporting this study and its findings are available within the article, its Supplementary Information and associated files. All source data for this study have been deposited in [https://www.scidb.cn/en/s/ueEBre] or are available from the corresponding author upon request. Source data are provided with this paper.

## Code availability

The codes supporting the findings of this study are available from the corresponding authors upon request.

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

## Acknowledgements

This work was supported by the National Key R&D Program of China (2023YFB4502200), National Natural Science Foundation of China (61925401, 92064004, 61927901, 8206100486, 92164302), Beijing Natural Science Foundation (L234026) and the 111 Project (B18001).

## Author contributions

K.Y. and Y.W. contributed equally to this work. R.Y., C.L., G.L., and C.G. fabricated the VO2 devices. K.Y., Y.W., and T.Z. performed characterization and experimentation. Y.W., P.J.T., T.Z., C.W., X.Z., and S.W. performed the simulations. K.Y., Y.W., R.H., T.Z., and Y.Y. prepared the manuscript. T.Z. and Y.Y. directed all the research. All authors analyzed the results and implications and commented on the manuscript at all stages.

## Competing interests

The authors declare no competing interests.
