## [Peer Review File · Nature Communications]

REVIEWER COMMENTS

Reviewer #1 (Remarks to the Author):

High-order sensory processing nanocircuit based on coupled VO2 oscillators.

Authors aim to emulate the biological neural networks that convert sensory information into periodic spiking activities as in visual and haptic sensory. It is proposed to utilize the dynamics of coupled VO2 oscillators to convert the sensory signals into oscillations. This is a new approach for the current state of the art on using coupled VO2 oscillators for pattern recognition and solving combinatorial optimization problems.

There are several concerns with the manuscript:

1. The core contribution of the paper is not clear. For example, there is no definition of the sensory patterns, what they are, in terms of frequency and amplitude. The collective dynamics of coupled oscillators when initialized with external stimuli of random frequencies, there is no mathematical derivation that the network will stabilize and provide stationary phase patterns. Thus, the mathematical link from natural frequencies (f_1, f_2, \dots, f_n) to $(O_1(t), o_2(t), \dots, O_n(t))$ is missing.
2. It is stated that output $((O_1(t), o_2(t), \dots, O_n(t))$ is transmitted to the decision-making network for further analysis and classification. Three equations are listed to describe the decision making model based on mean-field approach. The mathematic link from how $((O_1(t), o_2(t), \dots, O_n(t))$ relate to these equations is missing. What are these equations solving for and where is the input from the first network introduced?
3. Overall, the mathematical foundation of how to process sensor signals with coupled oscillators is not clearly explained to assess the contribution of the paper. A lot of the terms are not clearly defined.
4. Fig 1.c and Fig.3a give an illustration of the concept but there is no explanation of the mathematical derivation of sensor processing is computed.
5. In addition, the paper is not well written and there are lots of term that are not explained and not even clear why they brought up in the text, making it hard to read. For example
 - Page 3, what the definitions for ‘environmental variable”, “dynamic sensory solution”, ‘stronger power efficiency’, ‘portability’, ‘low-level sensory port’,
 - Page 4, no definition of in-sensory computing, what it entails and how that works, what current methods are, is lacking. Then the transition from the first paragraph to the second doesn’t fit.

- Several statements would require citation to back up. Such as 'Instead of extracting feature layer by layer in a hierarchical feedforward style, the retinal network in this shortcut adopts a structure combining two dynamic networks'.

- NMDA is not defined

6. A gesture recognition experimental implementation is provided and also shown in Fig.6. The inputs are more stretch sensors connected directly to VO2 devices coupled capacitively. This is quite confusing how the output voltages in 10V from stretch sensors generated from finger movements are suitable for biasing VO2 devices that are not in the same voltage range. Additionally, any random initialization will give a random phase output, so it can be insightful to show initial signals and their synchronisation. Why are the signal waveforms so different in Fig.6? What are the figures showing Response and Simulation time showing? What is response mean here and how is the classification of gesture done based on this response?

7. VO2 devices are known to suffer from low yield and variability. How is the non-uniformity of devices taken into account for signal processing? Would this be detrimental or advantageous?

8. Overall, paper writing can be enhanced for clarity and conciseness.

Reviewer #2 (Remarks to the Author):

The paper presents a capacitance-coupled oscillatory network based on VO2 memristive oscillators to emulate sensory processing. Here are my comments for the paper:

1. Abstract is too general. It does not provide relevant and specific information regarding the presented work.

2. Conventionally, the term sensor is used in two different ways in this field of research as follows: a) A sensing device is called a sensor. b) A sensing device/element with processing units is also called as sensor. Starting the abstract with statement, i.e., "In-sensor computing systems can process physical signals in situ with efficient encoding" create the confusion. Are authors referring case (a) or (b)? If it is case (a), then VO2 is not used as sensing element in the study and calling it in-sensor is not appropriate. If it is case (b), then use of in-situ will not be appropriate. Here, as far as I understand, VO2 device is used as a computing element while capacitor is used for sensing.

3. The terminology of memristive oscillatory neuron is confusing. Generally memristors are synapses and neurons are a separate unit. Why is the neuron memristive here?

4. All the figures are schematics. There is no microscope or SEM image of the device in the main figures. Even supplementary Fig. 1 looks heavily edited. Please provide clear microscope and SEM images.

5. VO2 device requires high current in several mAs and applied voltage in several volts. Authors didn't present any comparative analysis related to energy efficiency in comparison to other approaches.
6. It is not clear how many devices were fabricated and characterized.
7. Authors mentioned the existence of coupling among sensory nodes. However, authors don't provide any experimental proof of it other than simulation results.
8. It is not clear why the authors used normalized value of the capacitor voltages. They should report actual values to operate the system.
9. Paper Fig2. and supplementary Fig.2 results are not consistent with each other. In paper Fig2., Frequency is 300KHZ at 1000pF whereas in supplementary fig.2 300 KHz is at 30uF. It shows that used pressure sensor's characteristic values are not in alignment with the simulations in Fig2.
10. Can the authors provide an actual image of their experiment setup and used components with explanatory schematics and block diagrams?
11. Authors used the words, i.e., experimental, demonstration etc., at several places for the simulation setup. Authors should use these with actual experimental results.
12. Can authors discuss the integration details of pressure sensing device with mott device?
13. Considering the simplicity of the task (differentiating three gestures, or pressure), there are many other methods to achieve the same thing. The advantage of this scheme is not clear.

Overall, the Paper is not based on a sound rationale and does not present advances over existing methods. Spiking neurons based on VO2 have been extensively published by many groups in the past. This paper seems to be an application of them. More importantly the paper lacks relevant experimental results and analysis. I do not recommend its publication in Nature Communications.

Reviewer #3 (Remarks to the Author):

Dear Authors,

thanks for the interesting research. In this work, you proposed a gesture recognition based on a decision making algorithm and the properties of oscillating systems based on VO2 MIT switching devices. Concepts are easy to be understood and the application is well chosen for the neuron you propose. I suggest you, to have a stronger claim and impact, to better characterize the device part, which is a bit sacrificed, and highlight more the benefits of your architecture. In particular:

- 1) The device you proposed has the same electrical response of the devices reported in the literature, as in [39]. Which is the impact of the compliance current? Is it applied to protect the device? Did you see some geometrical factor? Is it a bipolar or unipolar device?

2) The quasi-static IV curve has a voltage window between 0 and 2V. The devices are then used in faster dynamics. There is no characterization of the device for pulse stimuli. I suggest you to better study and present the electrical response. For many memristive technologies, as volatile Ag-based RRAMs and Ovonic-threshold switching technologies, when the dynamic is fast the voltage required for the switching increases. Did you see similar effects?

3) Since the hysteresis present in the IV curve, is there a retention behavior of the switching? Which is the switching time? I suggest to characterize the switching in pulsed dynamic and better highlight the switching time and the retention properties, which limit the overall working frequency.

4) The quasi-static IV curve shows large currents in the range of mA. How did you compute the overall pulse energy?

Regarding the application:

5) there are several papers which propose gesture recognition using memristive architecture, with more complex architecture and more computational power. A comparison with such works may better highlight your claim.

Zhong, Y., Tang, J., Li, X. et al. A memristor-based analogue reservoir computing system for real-time and power-efficient signal processing. *Nat Electron* 5, 672–681 (2022). <https://doi.org/10.1038/s41928-022-00838-3>

B. Chen, J. Yao, J. Xia, R. Yang and X. Miao, "A Strain-Sensitive Flexible MoTe₂-Based Memristor for Gesture Recognition," in *IEEE Electron Device Letters*, vol. 44, no. 4, pp. 622-625, April 2023, doi: 10.1109/LED.2023.3249340.

1. Ceolini, E. et al. "Hand-Gesture Recognition Based on EMG and Event-Based Camera Sensor Fusion: A Benchmark in Neuromorphic Computing". *Frontiers in Neuroscience* vol. 14 (2020). <https://doi.org/10.3389/fnins.2020.00637>

6) The decision making module, as well as the force learning, should be explained more in detail, to provide a better comparison for future work and comparison. Which kind of architecture are? How many layers or building-blocks? Are there some references? Supplementary figure 7 is not explained, and the variable inside neither. In the energy consideration it should be included the also the second layer (force learning or decision making).

7) The application is based on 3 coupled neurons. What happens for larger system? Why didn't you exploit the full set of 8 different gestures you can encode (as in Supplementary figure 9)? Is it possible to use a simple synaptic layer (made by other memristive devices) to distinguish the different gestures?

8) How many attempts did you perform for each gesture? Did you consider also some cycle-to-cycle and device-to-device variability?

9) Supplementary figure 6 is not cited in the main text

MS No: NCOMMS-23-18953

Title: High-order sensory processing nanocircuit based on coupled VO₂ oscillators

Comments from Reviewer #1

Authors aim to emulate the biological neural networks that convert sensory information into periodic spiking activities as in visual and haptic sensory. It is proposed to utilize the dynamics of coupled VO₂ oscillators to convert the sensory

signals into oscillations. This is a new approach for the current state of the art on using coupled VO2 oscillators for pattern recognition and solving combinatorial optimization problems.

Our response: We would like to thank the reviewer for the professional suggestions from a theoretical view and the positive remarks on the novelty of this work. In this revised manuscript, we have meticulously addressed all the points raised and conducted additional experiments, modeling, and simulations (Fig.1d, Fig.3a, Fig. 5, Supplementary Fig. 4, 5, 11, 16-18, Equation 1-9) to strengthen the underlying rationale.

Besides, we have done a major revision of the entire article and removed the terms that may cause confusion to enhance the manuscript's clarity and comprehensibility. Our detailed responses to the comments and corresponding changes are shown as follows.

There are several concerns with the manuscript:

1. The core contribution of the paper is not clear. For example, there is no definition of the sensory patterns, what they are, in terms of frequency and amplitude. The collective dynamics of coupled oscillators when initialized with external stimuli of random frequencies, there is no mathematical derivation that the network will stabilize and provide stationary phase patterns. Thus, the mathematical link from natural frequencies (f_1, f_2, \dots, f_n) to ($O_1(t), o_2(t), \dots, O_n(t)$) is missing.

Our response: We would like to thank the reviewer for the constructive comments. In this work, we designed and experimentally demonstrated a dynamic oscillator-based computing architecture based on VO₂ memristive oscillators to solve sensory classification problems with low Energy-Delay Product (EDP). The system is tailored to harness the unique characteristics of VO₂ memristive oscillators, which exhibit high cycle-to-cycle uniformity and effectively mitigate device-to-device variation. Additionally, we have incorporated a decision-making module to offer a compact solution for post-processing the coupled oscillation signals.

Sensory pattern serves as an abstract representation of the physical stimuli received by neurons in the Oscillation Neural Network (ONN). For example, in a touch recognition task, a sensory pattern can be represented as $(1, 0, \dots, 1)$, where 1 signifies touched (high capacitance) and 0 signifies untouched (low capacitance). Different values of sensing capacitors will cause different intrinsic frequencies (f_1, f_2, \dots, f_n) of individual oscillators at a certain voltage V_{dd} . The circuit diagram is shown in the newly added **Fig. 1d**. While these oscillators are electrically coupled through coupling capacitors, the coupling network will stabilize and encode mode with synchronous output frequency f but different phases ($O_1(t), O_2(t), \dots, O_n(t)$). We can supplement the mathematical derivation process to establish a comprehensive mathematical framework and elucidate how the network achieves stabilization, leading to stationary phase patterns.

To address this question, we have revised the following sentences into Page 7 of the revised manuscript:

“As shown in **Fig. 1c**, the input sensory stimuli alter the sensing capacitance and determine the intrinsic frequencies of the VO₂ oscillators. As a result, the specific sensory pattern is converted to a set of natural frequencies (f_1, f_2, \dots, f_n) that are involved in the dynamic evolution of the network. The oscillatory neurons with the natural frequencies (f_1, f_2, \dots, f_n) interact with each other and evolve towards a collective ground state which refers to the synchronization with stationary phase pattern ($o_1(t), o_2(t), \dots, o_n(t)$) (**Fig. 1b**). Here, s_n serves as abstract representation of the physical stimuli received by neuron n in the ONN, while $f_n, o_n(t)$ denotes the natural frequency and voltage output of neuron n in the ONN.”

The VO₂ oscillators with different natural frequencies can be coupled and output signals with the same frequency but different phases. It is inspired by a theoretical model of the Kuramoto oscillator model, which is a sinusoidal oscillator that interacts via “weak” linear phase coupling. The coupling Kuramoto oscillators can be described by the equation (1):

$$\dot{\theta}_i = \omega_i + \frac{K}{N} \sum_{j=1}^N \sin(\theta_j - \theta_i), \quad i = 1, \dots, N \quad (1)$$

Where θ_i and ω_i are the phase and frequency of the i th oscillators. These oscillators can be weak-coupled with each other and the coupling affects only the phases without disturbing the frequency.

Specifically, as for VO₂ devices, we can set a VO₂ theoretical model to do circuit simulation from the circuit simulation level. According to experimental measurement as shown in **Figure 2c**, the VO₂ device is a threshold device that has four characteristic

parameters of V_{th} , V_{hold} , R_{on} , R_{off} ($R_{off} \gg R_{on}$). When the applied voltage exceeds the V_{th} , the device will rapidly change from the high resistance state (R_{off}) to low resistance (R_{on}). When the applied voltage below V_{hold} in the scanning back period, the device will also rapidly change from the low resistance state (R_{on}) to low resistance (R_{off}). The corresponding physical dynamic is Mott transition.

To address this question, we have revised the following sentences into Page 11-12 of the revised manuscript:

“When the VO₂ device is applied with a bi-directional quasi-static voltage sweep, the corresponding current displays a hysteresis switching response (**Fig. 2c**). Specifically, the device initially stays in a semiconducting monoclinic state, a high resistive state (HRS, R_{off}). When the voltage across exceeds a threshold value, denote as V_{th} , the metal-insulator transition (MIT) takes place and VO₂ turns to a metallic rutile phase, a low resistive state (LRS, R_{on}). An abrupt increase in the current can be observed which also exhibits a saturation in our case due to the compliance during measurement. Due to the orders of magnitude change in resistance, the compliance current is applied to protect the devices. The quasi-static voltage sweep with no compliance is shown in Supplementary Fig. 2. In the scanning back period, when the voltage drops below the critical value for maintaining the metallic state, denoted as V_{hold} , the VO₂ layer undergoes a reverse process and returns to its high resistive state which corresponds to the sharp decrease in current. The buried physics in MIT of the VO₂ device has attracted wide interest yet⁴⁰⁻⁴¹. The switching of VO₂ is believed to involve complex electronic and structural phase transitions⁴². Taking advantage of this strong nonlinear transition

in the VO₂ device, vital behaviors of biological neurons can be emulated and more nonlinear coupling behaviors can be constructed further.”

Then we connect the VO₂ devices with resistance and sensing capacitance as shown in Figure 2a, we have the following ordinary differential equation:

$$C_s \frac{dV_1}{dt} = \frac{V_{dd} - V_1}{R_L} - \frac{V_1}{R_{VO_2}} \quad (2)$$

where C_s is the sensing capacitance in parallel to the VO₂ device. V_m is the output voltage across the VO₂ device. The VO₂ resistance is $R_{VO_2} = R_{off}$ in HRS and $R_{VO_2} = R_{on}$ in LRS. For simplicity, we assume that R_{off} and R_{on} are constant in our analyses (e.g. $V_{th} = 3.5$ V, $V_{hold} = 1.5$ V, $R_{off} = 20$ kohm, $R_{on} = 400$ ohm). The equation indicates that the device can oscillate under appropriate loads (e.g. $C = 500$ pF, $R = 4$ Kohm, $V_{dd} = 5$ V). Different sensing capacitance will cause different oscillators' intrinsic oscillation frequencies (f_1, f_2, \dots, f_n).

To address this question, we have revised the following sentences into Page 18-20 of the revised manuscript:

“The VO₂ device is connected with resistance and sensing capacitance to form an oscillation circuit, as shown in **Fig. 2a**. Under a constant voltage input V_{dd} , the VO₂ device is expected to repetitively transfer between its HRS and LRS and an oscillating signal can be obtained at the output node. The VO₂ oscillator circuit can be described as the following ordinary differential equation:

$$C_s \frac{dV_1}{dt} = \frac{V_{dd} - V_1}{R_L} - \frac{V_1}{R_{VO_2}} \quad (2)$$

where C_s is the sensing capacitance in parallel to the VO₂ device. V_1 is the output

voltage across the VO₂ device. Specifically, the initial high resistance of the VO₂ will lead to an accumulated charge of capacitance until the device's voltage arrives V_{th} . It allows the transition towards the metal phase to take place. Subsequently, due to the discharge of capacitors, when the devices's voltage is lower than V_{hold} , the low resistance will no longer get sufficient voltage to hold the metal state and the device will go back to its HRS. It should be pointed out that the constraint of the series resistance is implied above to offer a suitable working voltage range for the VO₂ device, that is a voltage higher than the V_{th} at its HRS and a voltage lower than its V_{hold} at its LRS. A phase plane analysis is shown in **Fig. 2d** to present the VO₂ voltage oscillation from the initial state under different parallel capacitors. It can be seen the VO₂ oscillation has a stable periodic orbit and any other initial state will fall into the stable orbit eventually."

Furthermore, we consider the coupling oscillators. We connect the oscillators with coupling capacitors as shown in Figure 1d. Then we can have the following ordinary differential equations:

$$C_{s1} \frac{dV_1}{dt} = \frac{V_{dd} - V_1}{R_L} - \frac{V_1}{R_{VO_21}} + C_{p12} \frac{d(V_2 - V_1)}{dt} + C_{p13} \frac{d(V_3 - V_1)}{dt}$$

$$C_{s2} \frac{dV_2}{dt} = \frac{V_{dd} - V_2}{R_L} - \frac{V_2}{R_{VO_22}} + C_{p23} \frac{d(V_3 - V_2)}{dt} + C_{p12} \frac{d(V_1 - V_2)}{dt}$$

$$C_{s3} \frac{dV_3}{dt} = \frac{V_{dd} - V_3}{R_L} - \frac{V_3}{R_{VO_23}} + C_{p13} \frac{d(V_1 - V_3)}{dt} + C_{p23} \frac{d(V_2 - V_3)}{dt}$$

These are the equations of coupling VO₂ oscillators. This equation implies the synchronization under proper parameters. With appropriate coupling capacitances, the oscillators with different intrinsic oscillation frequencies can oscillate with the same

frequency but in different phases. When we change the sensing capacitors, the system's phase different mode will also change, which encodes the sensing information. The time-dependent output V_i is the $(O_1(t), O_2(t), \dots, O_n(t))$.

To address this question, we have revised the following sentences into Page 19 and page 22 of the revised manuscript:

Page 19,

“A Flow chart of the coupled VO₂ memristive oscillators (in experiment), the decision-making module(in software), and the mathematical link between them is shown in **Fig. 3a**. We can have the following ordinary differential equations to describe the coupled oscillators:

$$C_{s1} \frac{dV_1}{dt} = \frac{V_{dd} - V_1}{R_L} - \frac{V_1}{R_{VO_21}} + C_{p12} \frac{d(V_2 - V_1)}{dt}$$

$$C_{s2} \frac{dV_2}{dt} = \frac{V_{dd} - V_2}{R_L} - \frac{V_2}{R_{VO_22}} + C_{p12} \frac{d(V_1 - V_2)}{dt}$$

These equations imply the synchronization under proper parameters. With appropriate coupling capacitances, the oscillators with different intrinsic oscillation frequencies can oscillate with the same frequency but in different phases. When we change the sensing capacitors, the system's phase different mode will also change, which encodes the sensing information. ”

Page 22,

“Then we can have the following ordinary differential equations:

$$C_{s1} \frac{dV_{thu}}{dt} = \frac{V_{dd} - V_{thu}}{R_{VO_21}} - \frac{V_{thu}}{R_{thu}} + C_{p12} \frac{d(V_{for} - V_{thu})}{dt} + C_{p13} \frac{d(V_{mid} - V_{thu})}{dt} \quad (7)$$

$$C_{s2} \frac{dV_{for}}{dt} = \frac{V_{dd} - V_{for}}{R_{VO_22}} - \frac{V_{for}}{R_{for}} + C_{p23} \frac{d(V_{thu} - V_{for})}{dt} + C_{p12} \frac{d(V_{mid} - V_{for})}{dt} \quad (8)$$

$$C_{s3} \frac{dV_{mid}}{dt} = \frac{V_{dd} - V_{mid}}{R_{VO_23}} - \frac{V_{mid}}{R_{mid}} + C_{p13} \frac{d(V_{thu} - V_{mid})}{dt} + C_{p23} \frac{d(V_{for} - V_{mid})}{dt} \quad (9)$$

These three neurons are denoted as n_{thu} , n_{for} and n_{mid} with their output V_{thu} , V_{for} and V_{mid} , accordingly. The sensory neurons are powered by the (10 V, 300 μ s) long voltage pulse as V_{dd} , and the series resistors are configured as $R_{thu} = R_{for} = R_{mid} = 8.2\text{k}\Omega$. The coupling capacitors of 100 pF are adopted.”

Fig. 1 The sensory processing framework in different systems. **a** Biological sensory system. Receptors and neurons convert physical stimuli into electrical pulses which are transmitted to the cortex for high-level processing. **b** Conventional sensory system. In digital processing way, the signals from sensors are first digitalized through ADC and then put into separate memory units and processing units. In the ANN way, a crossbar composed of sensors can perform linear operations and realize classification. **c** Dynamic sensory system. The proposed sensory system consists of a memristive oscillatory network and a decision-making module, converting discrete oscillator frequencies into phase differences at the same frequency through coupling. **d** Hardware mapping of the dynamic sensory system. The specific circuit diagram can encode the sensory information in the phase pattern after electrical coupling.

To verify whether the experimental measurement data is mathematically coupled, we further added the fast Fourier transform (FFT) results of three coupled VO₂ oscillators as shown in Supplementary Figure 18. The FFT results shows the consistency of the frequencies in all eight cases that 3 coupled oscillators can encode.

Supplementary Figure 18. The FFT result of three coupled sensory VO₂ oscillators

in all eight cases. It shows the consistency of frequencies.

2. It is stated that output $((O_1(t), o_2(t), \dots, O_n(t))$ is transmitted to the decision-making network for further analysis and classification. Three equations are listed to describe the decision-making model based on mean-field approach. The mathematic link from how $((O_1(t), O_2(t), \dots, O_n(t))$ relate to these equations is missing. What are these equations solving for and where is the input from the first network introduced?

Our response: We would like to thank the reviewer for the constructive comments. As shown in Figure 1d, the outputs $((O_1(t), O_2(t), \dots, O_n(t))$ are time-continuous signals with irregular waveforms from VO_2 oscillators and need special treatment for further analysis. The conventional approach is to use a recurrent neural network, requiring a large number of hidden layer neurons. Here, we adopt a more efficient and compact network biologically processing oscillatory information — the decision-making module. The decision-making module can convert oscillation signals with the same frequency but different phases into the amplitudes of neurons as the classification results. It has the characteristic of winner-takes-all and is mathematically interpretable (*J. Neurosci.*, 26, 1314-1328 (2006)).

The three equations are listed below. They describe the behavior of each neuron in the decision-making module. We added a schematic in **Fig. 3a** and **Supplementary Figure 13** to help readers better understand the post-processing procedures.

$$x_i(t) = J_E s_i + \sum_{j \neq i}^{N_{dm}} J_M s_j + Z_i + I_0 \quad (1)$$

$$r_i = \frac{\beta}{\gamma} \ln \left[1 + e^{\frac{x_i - \theta}{\alpha}} \right] \quad (2)$$

$$\tau_s \frac{ds_i}{dt} = -s_i + \gamma(1 - s_i)r_i \quad (3)$$

where x_i and r_i denotes the synaptic input and neuronal activity of the i th neuron. s_i represents the synaptic current due to receptors. The equation (1) describes the neurons in the decision-making module that receive synaptic current from themselves feedback $J_E s_i$, other neuron $\sum_{j \neq i}^{N_{dm}} J_M s_j$ and the input from the former layer Z_i . The equation (2) describes the neuron's nonlinear activation function with a threshold θ . The equation (3) describes the slow dynamics of the synaptic current.

The first network's output is the voltage oscillation signal \mathbf{V} , which is the $((O1(t), O2(t), \dots, On(t)))$ in **Fig. 1c-d**. First, the signal \mathbf{V} through linear combination converts to \mathbf{Z} . The impact of the linear combination is to adjust the signal to an appropriate range. The weight of the linear combination can be trained by Force learning (*Neural Netw.*, **143**, 74-87 (2021)). Then \mathbf{Z} is the input to the decision-making module as shown in Equation 1. The expected output \mathbf{r} of the decision-making module is that one neuron fires with maximum strength while the others fire less.

To address this question, we have revised the following sentences into Page 18-20 of the revised manuscript:

“The decision-making network we employed is a simplified mean-field decision-making model³⁴⁻³⁵. In the decision-making network illustrated in **Fig. 1c**, there are n neurons each representing one of the classification decisions of the sensory pattern.

Among the neurons, mutual inhibition is introduced to make them compete with each other and determine the only winner as the final result. The model can be mathematically described by the following equations³⁵:

$$x_i(t) = J_E s_i + \sum_{j \neq i}^{N_{dm}} J_M s_j + Z_i + I_0 \quad (1)$$

$$r_i = \frac{\beta}{\gamma} \ln \left[1 + e^{\frac{x_i - \theta}{\alpha}} \right] \quad (2)$$

$$\tau_s \frac{ds_i}{dt} = -s_i + \gamma(1 - s_i)r_i \quad (3)$$

where x_i and r_i denotes the synaptic input and neuronal activity of the i_{th} neuron. s_i represents the synaptic current due to receptors. The equation (1) describes the neurons in the decision-making module that receive synaptic current from themselves feedback $J_E s_i$, other neuron $\sum_{j \neq i}^{N_{dm}} J_M s_j$ and the input from the former layer Z_i . The equation (2) describes the neuron's nonlinear activation function with a threshold θ . The equation (3) describes the slow dynamics of the synaptic current. Specifically, the synaptic input x_i described by Equation 1 is composed of three parts, which are the self-excitation J_{ES_i} ($J_E \geq 0$), total recurrent input from other neurons with $J_M \leq 0$ indicating mutual inhibition and the feedforward input I_i from the ONN module. The neuronal activity r_i can be further calculated by x_i using the nonlinear activation function in Equation 2 with parameters α , β , γ and threshold θ . The slow dynamics of the synaptic current originated from the activity-dependent NMDA receptors is formulated in Equation 3, which plays a crucial role in the spatio-temporal information processing of the decision-making network. The time constant τ_s ($\gg 1$) in Equation 3 controls the time window for integrating input over time by the decision-making neurons. The first coupled oscillation network's output is the voltage

oscillation signal V , which is the $((O1(t), O2(t), \dots, On(t)))$ in **Fig. 1c-d**. First, the signal V through linear combination converts to Z . The impact of the linear combination is to adjust the signal to an appropriate range. The weight of the linear combination can be trained by Force learning³⁵. Then Z is the input to the decision-making module as shown in Equation 1. The expected output r of the decision-making module is that one neuron fires with maximum strength while the others fire less.”

Fig. 3 Touch recognition with the bio-inspired sensory recognition system. a Flow chart of the coupled VO₂ memristive oscillators (in experiment), the decision-making module (in software), and the mathematical link between them. **b** The phase pattern without touch event. The outputs of the two neurons in ONN are not synchronized to the identical frequency. **c** The phase pattern with touch event. The outputs of the two neurons in ONN are locked in frequency with stable difference in phase. **d** Outputs of

the decision-making network corresponding to the non-synchronized condition (without touch). e Outputs of the decision-making network corresponding to the synchronization pattern (with touch).

3. Overall, the mathematical foundation of how to process sensor signals with coupled oscillators is not clearly explained to assess the contribution of the paper. A lot of the terms are not clearly defined.

Our response: We would like to thank the reviewer for the constructive comments. To clearly explain the coupled VO₂ oscillators, we provide a figure to illustrate how to process sensor signals with coupled oscillators in **Fig. 1d**, accompanied by the requisite mathematical derivations to establish a solid foundation. We have provided an elaborate explanation in response to Q1.

Given the interdisciplinary nature of this paper, we have diligently reviewed and defined terminology from various domains to enhance the comprehensibility of the manuscript.

4. Fig 1.c and Fig.3a give an illustration of the concept but there is no explanation of the mathematical derivation of sensor processing is computed.

Our response: We would like to thank the reviewer for the constructive comments. Initially, we built and demonstrated experimentally the coupling VO₂ oscillators for sensing processing, as shown in Supplementary Figure 11. However, we acknowledge

that the mathematical derivation was not included. We are now able to present the equation 1-9, Fig.1d, and Fig. 3a to better explain how the sensor processing is computed in mathematical derivation.

Supplementary Figure 11. A visual representation of the experiment setup, along with block diagrams of the components used.

5. In addition, the paper is not well written and there are lots of term that are not explained and not even clear why they brought up in the text, making it hard to read. For example

- Page 3, what the definitions for ‘environmental variable”, “dynamic sensory solution”, ‘stronger power efficiency’, ‘portability’, ‘low-level sensory port’,
- Page 4, no definition of in-sensory computing, what it entails and how that works, what current methods are, is lacking. Then the transition from the first paragraph

to the second doesn't fit.

- Several statements would require citation to back up. Such as 'Instead of extracting feature layer by layer in a hierarchical feedforward style, the retinal network in this shortcut adopts a structure combining two dynamic networks'.

- NMDA is not defined

Our response: We would like to thank the reviewer for the constructive comments.

We have removed the ambiguous terms and explained the terms that appear in the manuscript as much as possible to improve its readability.

For example, we remove terms like 'environmental variable', 'dynamic sensory solution', 'stronger power efficiency', 'portability', and 'low-level sensory port' to alleviate any potential confusion.

Furthermore, we have added information regarding in-sensory computing. The introductory paragraph sets forth a visionary concept of "letting devices' physics do the computing" (*Proc. IEEE*, **107**, 73-89 (2018)). This work delves into the interaction between the dynamics of the external world and the intrinsic physical attributes of the devices, facilitated through sensory units.

We have revised the introduction part to strengthen the logic and clarity of the manuscript.

We have added the citations on statements. Such as 'Instead of extracting feature layer by layer in a hierarchical feedforward style (*Nature*, 521, 436-444 (2015)), the retinal network in this shortcut adopts a structure combining two dynamic networks' (*J. Neurosci.*, 34, 13458-13471 (2014))

The NMDA (N-methyl-D-aspartate (NMDA) receptor) is the predominant molecular device for controlling synaptic plasticity and memory function. To avoid using too many confusing terms from other fields in the manuscript, we remove the words “NMDA”.

To address this question, we have revised the following sentences into Page 3 and Page 6 of the revised manuscript:

Page 3,

“The new technological innovations, including wearable electronics¹, auto-driving², and virtual reality³, are calling for advanced sensory systems, which should reduce the redundant data movement between sensors and processing units to provide improved area and energy efficiencies. In a general sensory system, the sensory data follows a hierarchical processing flow from low-level sensory processing (like encoding, filtering and feature enhancement) to high-level abstract representation (like recognition, classification, and localization)⁴. The biological sensory system adopts an efficient method to handle sensing in the noisy analog domain. The signals pass through skin receptors and afferent neurons for pre-processing and finally reach to spinal cord for post-processing (**Fig. 1a**).

With regards to low-level sensory processing, it’s vital to encode and transmit the proliferated data from the sensory nodes efficiently. However, the conventional architecture suffers from inefficient power consumption and notable latency. As shown in **Fig. 1b**, the detected data from the sensors must initially undergo digitization through an analog-to-digital (ADC) circuit before being temporarily stored in a memory unit,

awaiting processing by the computing unit. This process caused a great number of time delays and energy consumption. To tackle this problem, novel computing primitives such as near-sensor computing and in-sensor computing have been proposed and demonstrated on emerging nanoelectronics devices⁵⁻⁷. As for near-sensor computing, processing units or accelerators are located beside sensors and perform specific computational tasks at sensor endpoints. As for in-sensor computing, individual self-adaptive sensors or multiple connected sensors can directly pre-process sensory information. These innovative solutions effectively minimize redundant data transfers between sensors and external circuits, optimizing raw features in real-time and seamlessly executing vector-matrix multiplication for artificial neural networks.”

Page 6,

“Instead of extracting features layer by layer in a hierarchical feedforward style²⁸, the retinal network in this shortcut adopts a structure combining two dynamic networks²⁹”

6. A gesture recognition experimental implementation is provided and also shown in Fig.6. The inputs are more stretch sensors connected directly to VO2 devices coupled capacitively. This is quite confusing how the output voltages in 10V from stretch sensors generated from finger movements are suitable for biasing VO2 devices that are not in the same voltage range. Additionally, any random initialization will give a random phase output, so it can be insightful to show initial signals and their synchronization. Why are the signal waveforms so different in Fig.6? What are the figures showing Response and Simulation time showing?

What is response mean here and how is the classification of gesture done based on this response?

Our response: We would like to thank the reviewer for the constructive comments.

The stretch sensors essentially function as capacitors capable of varying their capacitance values in response to changes in stretch length. In our setup, we have connected the VO₂ devices in parallel with these stretch sensors, along with a series resistor, effectively forming a VO₂ oscillator with sensing capabilities, as depicted in **Fig. 1d**. The V_{dd} voltage is fixed at 10V, and the series resistor is set to 8.2k Ω . Variations in the sensing capacitance values lead to fluctuations in the frequency of the VO₂ oscillator, with an oscillation amplitude of approximately 3.5V, contingent upon the V_{th} threshold. When these VO₂ oscillators are interconnected, they generate a synchronization pattern that encodes information in the phase differences.

The random initialization will affect the phase difference after coupling, which is a crucial concern. In our experiments, we took the manual approach of ensuring that the initial charge on the capacitor was set to zero, thereby establishing stable initial state of the system. To address this, we can introduce a start-up circuit to regulate the initial voltage across each node, as shown in **Supplementary Figure 16**. The start-up circuit employs a voltage divider to determine the initial stage of the coupled oscillators. And we supplement the transient process at the beginning of coupling in **Supplementary Figure 17**. The transient process before stable coupling is around 20 μ s.

To address this question, we have revised the following sentences into Page 24 and

Supplementary Figure 16 of the revised manuscript:

“The random initialization will give a random phase pattern output. To ensure that the initial state doesn’t affect the phase difference after coupling, a start-up circuit can be set to make sure the phase modes of each output node remain stable in multiple experiments. The set-up circuit is a simple voltage divider circuit to determine the initial stage of the coupled oscillators. The total circuit diagram of ONN with a start-up circuit is shown in Supplementary Figure 8. We supplement the transient process at the beginning of coupling in Supplementary Figure 17 to show initial signals and their synchronization. The transient process before the stable coupling is around 20 us.”

Supplementary Figure 16. Start-up Circuit with ONN of three coupled oscillatory neurons with capacitive sensor. The Start-up circuit is to make sure the initial phase doesn’t affect the phase difference after coupling, keeping the initial state stability of the dynamic computing system.

Supplementary Figure 17. The transient coupling process of three coupled sensory VO₂ oscillators. The transient time is less than 20 us.

The signal waveforms look different and irregular. This is a major challenge for post-processing and we use the decision-making module to solve it. The irregularity stems from the time discrepancies between the charging and discharging of various capacitors. Besides, the coupled oscillators' frequency vary across different sensory cases. For the gesture “rock”(111), the frequency is 45kHz, while for “paper”(000), it is 91kHz.

The figures showing Response and Simulation time are all about the decision-making module, which is in software simulation for post-processing classification. They demonstrate that when irregular oscillation signals, where phase differences carry information, are input into the decision-making network, the network can produce

clear and distinguishable signals. In gesture recognition tasks, each neuron in the decision-making module corresponds to a specific gesture. When a gesture is encoded by the coupled VO₂ oscillators, the corresponding neuron in the decision-making module exhibits maximum output, effectively inhibiting other neurons. This transformation converts complex phase patterns into the firing strength of neurons, facilitating easy classification.

7. VO₂ devices are known to suffer from low yield and variability. How is the non-uniformity of devices taken into account for signal processing? Would this be detrimental or advantageous?

Our response: We would like to thank the reviewer for the constructive comments. In this work, the VO₂ thin film is large-area grown on c-Al₂O₃ substrate by the epitaxial method and we pattern metal electrodes with electron beam lithography (EBL) along with electron beam evaporation and lift-off. We fabricated more than 200 devices in total.

The variability of VO₂ devices contains cycle-to-cycle variation and device-to-device variation. The VO₂ devices and the coupling oscillation method can effectively address these two critical device-related challenges.

The high degree of cycle-to-cycle uniformity of VO₂ devices is an indispensable condition for this experiment. Compared to other threshold devices, our VO₂ devices exhibit remarkable consistency. We performed 200 times I-V sweeps on a single device.

The result is shown in **Supplementary Figure 4**. The deviation of the threshold voltage is 0.00021 while the deviation of the hold voltage is 0.00013. This low level of cycle-to-cycle variation is essential for the practical demonstration of oscillation, as the oscillator must consistently output a constant frequency.

To address this question, we have revised the following sentences into Page 13 of the revised manuscript:

“The monocrystalline VO₂ memristor provides low cycle-to-cycle variation (**Fig. 2c** and **Supplementary Fig. 4**). The deviation of the threshold voltage is 0.00021 and the deviation of the hold voltage is 0.00013. The low cycle-to-cycle variation is necessary for the practical oscillation demonstration because the oscillator must consistently output a constant frequency.”

Supplementary Figure 4. Measurement of VO₂ device cycle-to-cycle variation. (a) quasi-static voltage scanning of a VO₂ device (200 cycles). (b) The cycle-to-cycle variation of VO₂ device's threshold voltage and hold voltage.

For device-to-device variation, we conducted I-V sweeps on 40 devices. The result is shown in **Supplementary Figure 5**. The threshold voltage under quasi-static voltage

sweep varies from 3.55 V to 4.10 V and the hold voltage varies from 1.52 V to 1.96V. Devices with similar performance are more likely to establish out-of-phase synchronization after capacitive coupling.

To address this question, we have revised the following sentences into Page 13 of the revised manuscript:

“40 VO₂ devices are characterized to study their device-to-device variation, which will affect different natural frequencies of VO₂ oscillators, as shown in **Supplementary Fig. 5**. The threshold voltage under quasi-static voltage sweep varies from 3.55 V to 4.10 V and the hold voltage varies from 1.52 V to 1.96V. Devices with similar performance are more likely to establish out-of-phase synchronization after capacitive coupling.”

Supplementary Figure 5. Measurement of VO₂ device device-to-device variation in 40 devices.

Different VO₂ devices exhibit notable device-to-device variation even at the same size, which can be disadvantageous when used as individual oscillators. However, the

coupling oscillation strategy we adopt enables the synchronization of two oscillators with distinct intrinsic oscillation frequencies to a common frequency. Subsequently, we encode and classify them based on their phase patterns. The alteration in sensing capacitance exerts a more pronounced impact on the phase difference after the oscillators are coupled. Consequently, this strategy serves to the mitigation of device-to-device variation.

To address this question, we have revised the following sentences into Page 27 of the revised manuscript:

“The volatile memristive devices including VO₂, NbO₂ suffer from the large device-to-device variation. Different VO₂ devices exhibit notable device-to-device variation even at the same size, which can be disadvantageous when used as individual oscillators. However, the coupling oscillation strategy we adopt enables the synchronization of two oscillators with distinct intrinsic oscillation frequencies to a common frequency. Subsequently, we encode and classify them based on their phase patterns. The alteration in sensing capacitance exerts a more pronounced impact on the phase difference after the oscillators are coupled. Consequently, this strategy serves to the mitigation of device-to-device variation.”

8. Overall, paper writing can be enhanced for clarity and conciseness.

Our response: We would like to thank the reviewer for the constructive comments. This work spans across several domains, and your comments have aided us in providing a more comprehensive representation of our research. We have undertaken significant revisions to enhance the clarity and conciseness of the sensory computing process.

Comments from Reviewer #2

Overall, the Paper is not based on a sound rationale and does not present advances over existing methods. Spiking neurons based on VO₂ have been extensively published by many groups in the past. This paper seems to be an application of them. More importantly the paper lacks relevant experimental results and analysis. I do not recommend its publication in Nature Communications.

Our response: We would like to thank the reviewer's meticulous evaluation and comments on this paper. We acknowledge that some of the issues raised stem from potential ambiguity in our initial written expressions. For example, the misunderstanding regarding our coupling oscillation being based on software simulation, when in fact, all the data related to VO₂ oscillators' coupling oscillation is derived from real electrical measurements.

To address the challenge of demonstrating coupled oscillations in real experiments, we have fabricated high-performance monocrystalline VO₂ thin films and the VO₂ devices exhibiting high cycle-to-cycle uniformity, which is critical factor in advancing the practical application of memristive oscillators. Although individual VO₂ oscillator has been fabricated and studied by many groups before, achieving stable coupled oscillation in physical realization represents a significant advancement, particularly in the realm of dynamic computing.

Importantly, our work represents a pioneering effort in utilizing experimental

memristive coupling oscillators to achieve sensory information encoding. This underscores the computing potential of nonlinear oscillation circuits. In our study, we have meticulously analyzed the coupling dynamics and have further designed a decision-making module to classify the phase patterns of VO₂ oscillators, providing a comprehensive solution for dynamic sensory processing.

Our research has the potential to drive the development of memristive oscillators in constructing complex and dynamic nonlinear networks for future large-scale coupling oscillation applications. We kindly request the reviewer to reconsider our work in light of these clarifications. In this revised manuscript, with a focus on addressing the issues proposed, we have made the following key improvements and clarifications:

- 1、 To provide a sound rationale, we supplement the mathematical foundation of coupling oscillation behaviors of VO₂ memristive devices and provide a more detailed computing process description (Fig. 1d, Fig. 3a, Equation 1-9).

- 2、 In order to emphasize our contributions and the advantages of adopting the coupling oscillation method in sensing signal processing, we have included a comparison with prior works in the relevant field. (Supplementary Table 2)

- 3、 We have conducted a comparison to underscore the new dynamics for computing of the coupled oscillation architecture over existing methods using single spiking neurons. (Fig. 5, Supplementary Figure 18)

- 4、 To affirm that the data regarding coupling oscillation is derived from real experimental demonstrations, we have incorporated photos of the experiments,

including oscilloscope images. Furthermore, we have conducted additional electrical tests on the devices to provide a more comprehensive characterization.(Supplementary Figure 1-12)

5、 We have revised the abstract and the main body of the article, rectifying writing errors and elucidating terms that may be prone to ambiguity. These enhancements will help readers in achieving a better understanding of our work.

The point-to-point responses and changes made are listed below.

The paper presents a capacitance-coupled oscillatory network based on VO₂ memristive oscillators to emulate sensory processing. Here are my comments for the paper:

1. Abstract is too general. It does not provide relevant and specific information regarding the presented work. .

Our response: We would like to thank the reviewer for the constructive comments. To improve the clarity and effectiveness of our manuscript, we have revised the abstract to provide more relevant and specific information about the presented work. These changes will assist readers in better understanding the innovation and contributions of our manuscript.

To address this question, we have rewritten the abstract of the revised manuscript in page 2:

“Conventional circuit elements are constrained by limitations in area and power

efficiency at processing physical signals. Recently, researchers have delved into high-order dynamics and coupled oscillation dynamics utilizing Mott devices, revealing potent nonlinear computing capabilities. However, the intricate yet manageable population dynamics of multiple artificial sensory neurons with spatiotemporal coupling remain unexplored. Here, we present an experimental hardware demonstration featuring a capacitance-coupled VO₂ phase-change oscillatory network. This network serves as a continuous-time dynamic system for sensory pre-processing and encodes information in phase differences. Besides, a decision-making module for special post-processing through software simulation is designed to complete a bio-inspired dynamic sensory system. Our experiments provide compelling evidence that this transistor-free coupling network excels sensory processing tasks such as touch recognition and gesture recognition, achieving significant advantages of fewer devices and lower energy-delay-product when compared to conventional methods. This work paves the way towards an efficient and compact neuromorphic sensory system based on nano-scale nonlinear dynamics.”

2. Conventionally, the term sensor is used in two different ways in this field of research as follows: a) A sensing device is called a sensor. b) A sensing device/element with processing units is also called as sensor. Starting the abstract with statement, i.e., “In-sensor computing systems can process physical signals in situ with efficient encoding” create the confusion. Are authors referring case (a) or (b)? If it is case (a), then VO₂ is not used as sensing element in the study and

calling it in-sensor is not appropriate. If it is case (b), then use of in-situ will not be appropriate. Here, as far as I understand, VO₂ device is used as a computing element while capacitor is used for sensing.

Our response: We would like to thank the reviewer for pointing out the terms that may cause ambiguity in our manuscript. In light of your feedback, we have made the necessary clarifications.

The terms "in-sensor" and "in-situ" in our work may cause confusion to readers and are not suitable for describing our approach. In our work, we utilize sensing capacitors and VO₂ devices to construct sensory oscillation neurons. The oscillation dynamics rely on VO₂ threshold devices as well as the charge of capacitors, making both components essential. Therefore, we have chosen to refer to this as a "dynamic sensory system" because it leverages both the physical phase transition dynamics of VO₂ devices and the circuit dynamics to process sensing signals through coupled oscillation.

The total architecture including dynamic sensory processing and post-processing is inspired by the Reference (*Nat. Electron.*, **3**, 664-671. (2020)). Due to the discrete nature of the devices used in our experiment, our approach aligns more closely with the concept of near-sensor computing. This approach enhances the sensor/processor interface and minimizes the need for redundant data transfer during sensory processing.

Furthermore, our dynamic sensory system has the potential for expansion into in-sensor computing, as described in more detail in Question 12. VO₂ device can respond to external stimuli such as light and temperature, leading to alterations in the dynamics of

coupled oscillation. This versatility makes it suitable for various sensory applications that involve different types of stimuli.

To avoid this confusion, we revised the article and discussed in-sensor computing and near-sensor computing in the introduction part.

To address this question, we have revised the following sentences into Page 2 of the revised manuscript:

“To tackle this problem, novel computing primitives such as near-sensor computing and in-sensor computing have been proposed and demonstrated on emerging nanoelectronics devices⁵⁻⁷. As for near-sensor computing, processing units or accelerators are located beside sensors and perform specific computational tasks at sensor endpoints. As for in-sensor computing, individual self-adaptive sensors or multiple connected sensors can directly pre-process sensory information. These innovative solutions effectively minimize redundant data transfers between sensors and external circuits, optimizing raw features in real-time and seamlessly executes vector-matrix multiplication for artificial neural networks.”

3. The terminology of memristive oscillatory neuron is confusing. Generally memristors are synapses and neurons are a separate unit. Why is the neuron memristive here?

Our response: We would like to thank the reviewer for pointing out the term that may cause confusion. Nowadays memristors are usually used for synapses to imitate

biological synaptic behaviors and perform linear vector-matrix multiplication operations in artificial neural networks. Neurons in artificial neural networks are only nonlinear activation functions. However, dynamic memristors have a more powerful computational significance in nonlinear systems which this work wants to show and promote (*Nat. Rev. Mater.*, **7**, 575-591 (2022))[1].

Leon Chua first proposed “memristor” in 1971 and constructed complex nonlinear circuit theories. In his definition, synapses are locally passive memristors, and neurons are made of locally active memristors (*Nanotech.*, **24**, 383001 (2013))[2]. The locally active memristor can have a persistent dynamic when coupled with other passive devices, working on the edge of chaos. In recent years, the Mott phase transition devices like NbO₂, VO₂ have been experimentally demonstrated high-order neuronal dynamics based on Leon Chua’s local activity theory (*Nature*, **548**, 318-321 (2017); *Nature*, **548**, 318-321 (2017); *Nat. Commun.*, **9**, 4661 (2018); *Adv. Mater.*, 2205451 (2022)) [3-6].

These references explicitly state that they are memristive devices.

1. Kumar, S., Wang, X., Strachan, J. P., Yang, Y., & Lu, W. D. Dynamical memristors for higher-complexity neuromorphic computing. *Nat. Rev. Mater.*, **7**, 575-591 (2022).
2. Chua, L. Memristor, Hodgkin–Huxley, and edge of chaos. *Nanotech.*, **24**, 383001 (2013)
3. Kumar, S., Strachan, J. P., & Williams, R. S. Chaotic dynamics in nanoscale NbO₂ Mott memristors for analogue computing. *Nature*, **548**, 318-321 (2017).
4. Kumar, S., Williams, R. S., & Wang, Z. Third-order nanocircuit elements for neuromorphic engineering. *Nature*, **548**, 318-321 (2017).
5. Yi, W, et al. Biological plausibility and stochasticity in scalable VO₂ active memristor neurons. *Nat. Commun.*, **9**, 4661 (2018)

6. Brown, T. D., et al. Electro-Thermal Characterization of Dynamical VO₂ Memristors via Local Activity Modeling. *Adv. Mater.*, 2205451 (2022).

Therefore, it is reasonable to call them memristive neurons. This work utilizes the volatility and threshold characteristics of memristors to construct oscillatory neurons.

The VO₂ neuron can be described in memristor state equations (*Adv. Mater.*, 2205451 (2022)) [6]. We provide the experimental bidirectional quasi-static voltage scanning of the VO₂ device (10 cycles) in **Supplementary Figure 2**.

To minimize this potential confusion, we have revised the article and substituted all the terms of “memristive neuron” with “memristive oscillators”.

To address this question, we have revised the **Supplementary Figure 2** of the revised manuscript:

Supplementary Figure 2. Experimental bidirectional quasi-static voltage scanning of a VO₂ device (10 cycles).

4. All the figures are schematics. There is no microscope or SEM image of the device in the main figures. Even supplementary Fig. 1 looks heavily edited. Please provide clear microscope and SEM images.

Our response: We would like to thank the reviewer for the constructive comments regarding the figures. We acknowledge that there may have been misunderstandings arising from the initial representations, which were not fully refined. Firstly, it's crucial to clarify that all the oscillation data presented in our work are derived from actual experiments. To mitigate potential misunderstandings, We have revised the figures and included additional supporting evidence in the supplementary material. Additionally, we are prepared to provide raw data upon request from readers. Furthermore, we added a clear microscope (**Fig. R1**) and scanning electron microscope (SEM) images of the devices in the main figure. We have also made revisions to Supplementary Fig. 1 to improve clarity.

Fig. R1. The microscope image of the VO₂ devices.

To address this question, we have revised **Supplementary Figure 1** of the revised manuscript:

Supplementary Figure 1. Microstructural and compositional characterization of

VO₂ device. (a) Structure of a planar VO₂ device. (b-d) SEM mapping of VO₂ devices. (e) Cross-sectional STEM image and corresponding EDS mapping of O, V, Au, Ti elements in the device.

5. VO₂ device requires high current in several mAs and applied voltage in several volts. Authors didn't present any comparative analysis related to energy efficiency in comparison to other approaches.

Our response: We would like to thank the reviewer for the constructive comments. The reason we adopt VO₂ oscillators is their high cycle-to-cycle uniformity. For practical experimental demonstrations, it is crucial to achieve a highly uniform oscillation frequency to enable the reproducible encoding of information through phase differences. The cycle-to-cycle variation is shown in **Supplementary Figure 4**. The deviation of the threshold voltage is 0.00021 and the deviation of the hold voltage is 0.00013. The high cycle-to-cycle variation is necessary for the practical oscillation demonstration because the oscillator must consistently output a constant frequency.

While oscillators based on Mott materials, such as VO₂, hold significant computational significance, they share a common challenge of high current consumption in the milliamper (mA) range. VO₂ devices exhibit a high threshold voltage and a low on-state resistance (approximately 300 ohms), contributing to their relatively high power consumption. In our evaluation, the energy per spike of the VO₂ oscillators can reach 2 nJ/per spike when the V_{dd} is 6 V and oscillation frequency is 1 MHz, resulting in a substantial power consumption of about 2 mW. In our gesture recognition task, the

power consumption is measured at 7.56 mW for three coupled VO₂ oscillators. However, despite this relatively high power consumption, the system offers low latency, resulting in a low Energy-Delay Product (EDP) of approximately 3.07 pJ*s in gesture recognition tasks.. The EDP is the multiply of energy and time, which is the key comparative metric in neuromorphic computing systems. The delay time of coupled oscillation is defined as the time of the transient process from the initial state to the stable coupling state, which is only 20 μ s in this work (as shown in **Supplementary Figure 3**). A comparison table of gesture recognition is listed in **Supplementary Table 2** and will be further elucidated in response to **Q13**.

There are other threshold devices that can be used for oscillators like volatile Ag-based atomic-switching devices and the B-Te-based Ovonic devices, which exhibit lower power consumption (around fJ/spike) owing to small threshold voltage and high on-state resistances. However, these devices have primarily remained at the single-device level in terms of application and are mainly explored through simulations. Our work combines the dynamics of VO₂ devices (locally active memristor) with circuit dynamics (spatially coupling) and represents the first experimental demonstration of coupled oscillation for sensory processing.

To address this question, we have added **Supplementary Figure 4** and **Supplementary Figure 17** of the revised manuscript and revised the following sentences into Page 12:

“The monocrystalline VO₂ memristor provides high cycle-to-cycle variation (**Supplementary Fig. 4**). The deviation of the threshold voltage is 0.00021 and the

deviation of the hold voltage is 0.00013. The high cycle-to-cycle variation is necessary for the practical oscillation demonstration because the oscillator must consistently output a constant frequency.”

Supplementary Figure 4. Measurement of VO₂ device cycle-to-cycle variation. (a) quasi-static voltage scanning of a VO₂ device (200 cycles). (b) The cycle-to-cycle variation of VO₂ device’s threshold voltage and hold voltage.

Supplementary Figure 17. The transient coupling process of three coupled sensory VO₂ oscillators in actual experiment. The transient time is less than 20 us.

6. It is not clear how many devices were fabricated and characterized.

Our response: We would like to thank the reviewer for the constructive comments. The microscopic photo is now available in **Figure R1**, providing a view of approximately 1/4 of the substrate. It's important to note that we fabricated over two hundred devices for our study. This ample number of devices adequately supports our concept of utilizing emerging nano-devices to demonstrate coupled VO₂ oscillators for dynamic computing, in which only a few devices are required.

To address the issue of device-to-device variation, we conducted tests on 40 devices. The threshold voltage under quasi-static voltage sweep varies from 3.55 V to 4.10 V and the hold voltage varies from 1.52 V to 1.96V. Importantly, our proposed computing approach involving coupled oscillations effectively mitigates this device-to-device variation. This is achieved by encoding the differences in the oscillators' natural frequencies as a single coupled frequency with varying phases. The value of the coupling capacitor plays a pivotal role in controlling the phase difference, as exemplified in **Fig. 2j**, where anti-phase behavior is observed under large capacitor values.

To address this question, we have added **Supplementary Figure 5** and revised the following sentences into Page 12, Page 27, and Page 31:

“40 VO₂ devices are characterized to study their device-to-device variation, which will affect different natural frequencies of VO₂ oscillators, as shown in **Supplementary Figure 5**. The threshold voltage under quasi-static voltage sweep varies from 3.55 V to 4.10 V and the hold voltage varies from 1.52 V to 1.96V. Devices with similar performance are more likely to form out-of-phase synchronization after capacitive coupling.”

Supplementary Figure 5. Measurement of VO₂ device device-to-device variation in 40 devices.

“The volatile memristive devices including VO₂, NbO₂ suffer from the large device-to-device variation. Different VO₂ devices exhibit notable device-to-device variation even at the same size, which can be disadvantageous when used as individual oscillators.

However, the coupling oscillation strategy we adopt enables the synchronization of two oscillators with distinct intrinsic oscillation frequencies to a common frequency. Subsequently, we encode and classify them based on their phase patterns. The alteration in sensing capacitance exerts a more pronounced impact on the phase difference after the oscillators are coupled. Consequently, this strategy serves to the mitigation of device-to-device variation.”

Page 31,

“More than two hundred devices were fabricated.”

7. Authors mentioned the existence of coupling among sensory nodes. However, authors don't provide any experimental proof of it other than simulation results.

Our response: We would like to thank the reviewer for raising this question. We would like to clarify that all the coupling results presented in the article are derived from experimental electrical testing, and to further bolster the credibility of our experiments, we have included experimental photos of the oscilloscope displays. **Figure R2 and R3** provide clear visual evidence of coupling among sensory nodes, which shows oscillations with the same frequency but different phases.

Furthermore, we conducted experiments involving all eight coupling modes using three coupled oscillators, as shown in **Supplementary Figure 20**. To provide a comprehensive analysis of the data, we employed Fast Fourier Transform (FFT), as shown in **Supplementary Figure 18**, which reveals that the three voltage output signals

exhibit consistent frequencies. Additionally, the 3D phase plane analysis, presented in Fig. 5, demonstrates that the three voltage output signals can form a limit cycle, further confirming the existence of coupling.

Figure R2. Real oscilloscope photo of coupled oscillation of VO₂ oscillators with sensing capacitor.

Figure R3. Oscilloscope photo of coupled oscillation of VO₂ oscillators with sensing capacitor.

To address this question, we have revised the following sentences on Page 23 of the revised manuscript, as well as Page 25-26:

Page 23,

“For mathematical verification of the coupling among sensory nodes, the Fast Fourier Transform (FFT) is applied to analyze the frequency consistency, as shown in **Supplementary Fig. 18.**”

Supplementary Figure 18. The FFT result of three coupled sensory VO₂ oscillators in all eight cases.

Page 25-26,

“In this work, we experimentally demonstrate the three coupled VO₂ oscillators, and the eight encoding modes are shown in Fig.5. Due to coupling characteristics, the output voltage signals can form clear trajectories in phase space, which reflect the classifiable modes as shown in Fig.5 a, c, e, g, i, k, m, o. The trajectories and their projections on x-y, y-z, x-z planes present the dynamic of stable limit cycles, whose shapes reflect the phase differences of different oscillators. The eight kinds of limit cycles attach great importance to dynamic computing based on memristors. It shows the actual physical memristors with variations can be accurately controlled in population dynamics.”

Fig. 5 The trajectories in the phase plane of three coupled VO₂ oscillators in eight coupled oscillation modes and the decision-making classification results. **a** The experimental trajectory in the phase plane of case ‘000’. **b** The decision-making classification results in the software of case ‘000’. Besides, the experimental trajectories in the phase plane of case ‘100’, ‘010’, ‘001’, ‘110’, ‘011’, ‘101’, ‘111’ correspond to **c**, **e**, **g**, **i**, **k**, **m**, **o**. The decision-making classification results in software of case ‘100’, ‘010’, ‘001’, ‘110’, ‘011’, ‘101’, ‘111’ correspond to **d**, **f**, **h**, **j**, **l**, **n**, **p**. Three coupled VO₂ dynamic devices can encode eight classifiable sensory modes in phase space.

8. It is not clear why the authors used normalized value of the capacitor voltages.

They should report actual values to operate the system.

Our response: We would like to thank the reviewer for proposing this question. Initially, we utilized normalized values to emphasize the phase difference. However, we recognize that this approach may have led to some misunderstandings in interpreting the experiment. As a response, we have now revised all the figures to represent the data in actual voltage values.

9. Paper Fig2. and supplementary Fig.2 results are not consistent with each other.

In paper Fig2., Frequency is 300KHZ at 1000pF whereas in supply fig.2 300 KHz is at 30uF. It shows that used pressure sensor's characteristic values are not in alignment with the simulations in Fig2.

Our response: We would like to thank the reviewer for bringing this issue to our attention. Upon reviewing the raw data, we have identified the error in **Supplementary Fig. 2**. The correct unit for sensing capacitance should be pF rather than μF . It's important to clarify that all the oscillation data presented in **Fig. 2** and throughout the article is derived from actual experimental electrical measurements. pointing out the problem. We check the raw data. The unit of sensing capacitance is actually pF rather than μF in Supplementary Fig.2. All the oscillation data from **Fig.2** is through actual experimental electrical measurement. The main difference between these two figures comes from the use of different series resistor, which is 3.4 kohm in Figure 2 and 10.2 kohms in Supplementary Figure 2.

To address this question, we have revised the **Supplementary Figure 9** and **Supplementary Figure 10**:

Supplementary Figure 9. The electrical characteristics of the pressure sensor and

the corresponding sensory VO₂ oscillator. (a) The capacitance characteristic of the pressure sensor. **(b)** The typical waveform of the haptic sensory neuron with pressure sensor. **(c)** Single-sided amplitude spectrum though FFT of the waveform in b). l)

the capacitance characteristic of the stretching sensor regarding varied stretch lengths ΔL .

m) The relationship between the stretch lengths ΔL and the output frequency.

Supplementary Figure 10. The electrical characteristics of the stretching sensor

and the corresponding sensory VO₂ oscillator. (a) The capacitance characteristic of the stretching sensor regarding varied stretch lengths ΔL . **(b)** The relationship between the stretch lengths ΔL and the output frequency.

10. Can the authors provide an actual image of their experiment setup and used components with explanatory schematics and block diagrams?

Our response: We would like to thank the reviewer for the constructive comments. Here we provide actual images of our experiment setup and used components with explanatory schematics and block diagrams, as shown in **Figure R4, Supplementary Figure 11-12**. The VO₂ device (at the probe station) is connected to the capacitance and resistance (on the breadboard) through cables. This configuration enables us to demonstrate the coupling of three oscillators effectively. Looking ahead, as our research advances, we envision the possibility of packaging VO₂ devices and integrating them onto PCB boards. This would enable us to create more intricate coupling networks and explore a broader range of behaviors, further showcasing the potential of memristors in nonlinear dynamic computing in future work.

Figure R4. The overall experimental instruments.

To address this question, we have revised the **Supplementary Figure 11** and **Supplementary Figure 12**:

Supplementary Figure 11. A picture of the experiment setup and used components with block diagrams.

Supplementary Figure 12. A picture of experiment setup and used components with explanatory schematics.

11. Authors used the words, i.e., experimental, demonstration etc., at several places for the simulation setup. Authors should use these with actual experimental results.

Our response: We appreciate the reviewer's perspective and understand the importance of distinguishing between papers that rely solely on simulation results and those that provide actual experimental demonstrations. We fully acknowledge the value of concrete experimental evidence in scientific research.

In our work, we have prioritized demonstrating the critical coupling oscillation characteristics of memristive oscillators through real experiments. We have provided experimental verification of coupling oscillation in Question.7. All the results about coupling oscillation are derived from actual experiments.

Furthermore, we acknowledge that the decision-making module for processing the irregular oscillation signals with continuous time is presented in a simulation context, as clearly stated in the initial version of the article. In this revised version, we have taken your feedback into account and labeled each image as either experimental or simulated to provide full transparency regarding the nature of the results.

12. Can authors discuss the integration details of the pressure sensing device with

mott device?

Our response: We would like to thank the reviewer for the constructive comments. In the current iteration of our work, the sensing devices are connected to VO₂ devices using breadboards and cables, as depicted in Supplementary Figures 8-9. There is potential benefit for integrating VO₂ devices and sensing devices on a single substrate.

The pressure sensor array can be made of MXene on flexible substrates. MXene is a 2-dimensional metal carbide/nitride exhibiting conductivity changes in response to external pressure. Besides, VO₂ devices can also have certain sensing functions. environment temperature will change the threshold voltage of VO₂ devices, affecting the intrinsic oscillation frequency and further affect the phase difference after coupling. Thus, VO₂ devices inherently possess the capability to undertake certain sensing functions, such as detecting changes in light, stress, and temperature.

Furthermore, the Mott device (VO₂ and NbO₂) can be fabricated at relatively low temperatures (< 300 °C), making them compatible with 3D integration techniques. Currently, we are working on developing a three-dimensional integration solution that integrates Mott devices as the top sensing layer, non-volatile arrays as the middle computing layer, and transistors as the bottom control layer. This approach allows us to achieve monolithic integration with sensors and various other computing units, expanding the scope of potential applications.

To address this question, we have revised the **sentences in Page 31**:

“In the current iteration of our work, the sensing devices are connected to VO₂

devices using breadboards and cables, as depicted in Supplementary Figures 8-9. There is potential benefit for integrating VO₂ devices and sensing devices on a single substrate. The pressure sensor array can be made of MXene on flexible substrates⁵⁸. MXene is a 2-dimensional metal carbide/nitride exhibiting conductivity changes in response to external pressure. Besides, VO₂ devices can also have certain sensing functions. For example, environment temperature will change the threshold voltage of VO₂ devices, affecting the intrinsic oscillation frequency and further affect the phase difference after coupling. Thus, VO₂ devices inherently possess the capability to undertake certain sensing functions, such as detecting changes in light, stress, and temperature. Furthermore, the Mott device (VO₂ and NbO₂) can be fabricated at relatively low temperatures (< 300 °C), making them compatible with 3D integration techniques.”

13. Considering the simplicity of the task (differentiating three gestures, or pressure), there are many other methods to achieve the same thing. The advantage of this scheme is not clear.

Our response: We would like to thank the reviewer for the constructive comments. In response to your suggestions, we have made several important enhancements to our work:

1. **Expanded Task Classification:** We have increased the task classification scale from 3 to 8 categories, as illustrated in **Fig. 5**. This extension demonstrates that three coupled VO₂ oscillators can effectively encode 8 distinct modes.

2. **Comparison with Other Approaches:** We have included a detailed comparison of our approach to gesture classification with methods based on memristors and CMOS in **Supplementary Table 2**. This comparison highlights three key advantages:

1) **Mitigating Device Variation:** The proposed computing approach utilizing coupled oscillation effectively addresses the issue of large device-to-device variation. Differences in the oscillators' natural frequencies can be encoded as a single coupled frequency with varying phases, and the coupling capacitor is the key factor that determines the phase difference. In general, variations in device characteristics primarily result in fluctuations in oscillation frequencies. Nevertheless, within our specific application, the phase difference of coupled oscillations assumes a pivotal role. Fluctuations in the oscillation frequencies of distinct devices will be mitigated through capacitive coupling since they will ultimately synchronize to the same frequency.

2) **Power Efficiency:** While VO₂ oscillators have relatively high power consumption due to their high threshold voltage and low on-state resistance, we acknowledge that there is room for improvement in this regard. Furthermore, we have highlighted the short delay time in coupled oscillation, defined as the time for the transient process from the initial state to the stable coupling state, which is only 20 μs in our work (as demonstrated in **Supplementary Figure 17**).

3) **Scalability:** Crucially, our approach can scale effectively to larger systems. N

coupled VO₂ oscillators can encode 2ⁿ distinguishable modes, offering a significant advantage over other dynamic processing methods that require exponential device consumption as system size grows.

To address this question, we have revised the **Supplementary Table 2** and the **sentences in page 25**:

“Couple VO₂ oscillators can have stronger computing superiority compared to the other methods including non-volatile memristors and CMOS circuits. A detailed comparison table with previous works is shown in Supplementary Table 2. The volatile memristive devices including VO₂, NbO₂ suffer from the large device-to-device variation. The proposed computing approach of coupling oscillation can solve this problem because the differences of the oscillators’ natural frequencies can be encoded as a single coupled frequency but different phase. And the coupling capacitor can suppress the variation in phase difference which can be controlled in anti-phase under large capacitor values. Besides, due to the high threshold voltage and low on-state resistor, the VO₂ oscillators have a high power, which can be improved in the future. The delay time in coupled oscillation can be defined as the time of the transient process from the initial state to the stable coupling state, which is only 20 μs in this work (as shown in Supplementary Fig. 3). Most importantly, when expanding to larger systems, *n* coupled VO₂ oscillators can encode 2ⁿ distinguishable modes while the other dynamic processing methods require exponential device consumption.”

Supplementary Table 2. A comparison table of gesture recognition task with

previous works including memristors and CMOS.

	Zhong, Y. et al	Chen, B., et al	Lu L. et al	Ceolini, E. et al.	Rui, Y. et al	This work
Device	TiO _x	MoTe ₂	P3HT/PEO	CMOS	VO ₂	VO ₂
Computing style	Near-sensor computing	In-sensor computing	In-sensor computing	Near-sensor computing	Near-sensor computing	Near-sensor computing
Computing method	Reservoir computing	Artificial neural network	Artificial neural network	Spiking neural network	Spiking neural network	Coupled oscillation
Memristor type	Volatile	Non-volatile	Non-volatile	/	Volatile	Volatile
D2D variation aware	NO	NO	NO	/	NO	YES
Device number	24M+2048M(experiment)	500M (simulation)	5(experiment) *512*128*84 *3(simulation)	/	10R+5C+5M(experiment)	3R+3M+3C(experiment) 3 (simulation)
Power	22.2 uW	/	/	29.4 mW	/	7.68 mW
Time	1.6 ms	/	/	5.89 ms	/	20 us
Energy	13.32 nJ	/	/	173.2 uJ	/	153.6 nJ
EDP	21.31 pJ*s	/	/	1.02 uJ*s	/	3.07 pJ*s

*The EDP (Energy Delay Product) is the multiply of energy and time, which is the key comparative indicators in neuromorphic computing systems

Comments from Reviewer #3

Thanks for the interesting research. In this work, you proposed a gesture recognition based on a decision-making algorithm and the properties of oscillating systems based on VO₂ MIT switching devices. Concepts are easy to be understood

and the application is well chosen for the neuron you propose. I suggest you, to have a stronger claim and impact, to better characterize the device part, which is a bit sacrificed, and highlight more the benefits of your architecture. In particular:

Our response: We would like to sincerely thank the reviewer for the very detailed and constructive suggestions. Your suggestions inspire us a lot and help us to better present the work of coupled VO₂ oscillators. In this revised manuscript, we have carefully considered all the points and performed additional experiments, developed new models, and carried out simulations (Fig.1d, Fig. 3a, Fig.5, Supplementary Fig. 2-7, 14, 15, Supplementary Table 2) to more effectively highlight the benefits of the VO₂ MIT device and the overall dynamic sensory architecture. Our detailed responses to the comments and corresponding changes are shown as follows.

1) The device you proposed has the same electrical response of the devices reported in the literature, as in [39]. Which is the impact of the compliance current? Is it applied to protect the device? Did you see some geometrical factor? Is it a bipolar or unipolar device?

Our response: We would like to thank the reviewer for the constructive comments. Article [39] (*Adv. Mater.*, 2205294 (2022)) fully discussed the material design principles of VO₂ mott devices to better utilize thermodynamics and kinetics of electronic transitions to build next-generation neuromorphic computing. In our work, we grew 20 nm VO₂ thin film on c-Al₂O₃ substrates in an epitaxial manner and then

fabricated the devices with 400 nm length, which show high cycle-to-cycle uniformity to meet the proposed application requirements.

The compliance current is applied to protect the devices. In our devices, when the phase transition occurs, the current passing through the VO₂ device will change from 400 uA to about 20 mA in the 0 - 5 V quasi-static voltage scanning range, as shown in **Supplementary Figure 3**. After several cycles (about 10), the device will remain in a low resistance state and can not go back to the insulated state. However, in oscillation applications, the time of a large current passing through the VO₂ device is too short to damage the device and thus, the device shows robust endurance.

To address this question, we have revised the following sentences into Page 11 of the revised manuscript:

“Given the orders of magnitude change in resistance (from 20 Kohm to 400 ohm), the compliance current is applied to safeguard the devices. The quasi-static voltage sweep without compliance is shown in **Supplementary Fig. 2**.”

Supplementary Figure 2. Experimental quasi-static voltage scanning of a VO₂ device without compliance current. The maximum current through the device can reach to around 20 mA.

As for the geometrical factor, in this work, we fabricated two sizes of the VO₂ devices. One has 400 nm in length, 2 μm in width, and the VO₂ film thickness of 20nm, while the other has 200 nm in length, 2 μm in width, and the same VO₂ film thickness of 20nm. The change in device length will affect the V_{th} , V_{hold} , R_{on} , R_{off} . The IV sweep is shown in **Figure R5**. Specifically, the smaller device (200 nm) has smaller parameters of V_{th} , V_{hold} , R_{on} , and R_{off} , with V_{th} of 1.4 V and V_{hold} of 0.9V. However, the device-to-device variation of the smaller device poses challenges for stable coupling. The experimental demonstration of coupled oscillation is based on devices of 400 nm length. Therefore, we have revised all the device data to 400nm devices throughout the manuscript to ensure consistency in device performance as presented in the manuscript.

Figure R5. The VO₂ devices with smaller sizes have smaller electrical parameters

V_{th} , V_{hold} , R_{on} , R_{off} .

The VO₂ device is a bidirectional symmetric planar device. It is volatile because its

resistance will rapidly increase after removing the applied voltage (automatically from on-state to off-state). Consequently, there is no need for a RESET operation. The VO₂ device is actually a non-polar device. We provide the device's response to positive and negative voltage quasi-static scanning. The result is shown in **Supplementary Figure 3**. It shows the VO₂ device is a symmetric volatile memristive device.

To address this question, we have revised the following sentences into Page 11 of the revised manuscript:

“The VO₂ device is a bidirectional symmetric planar device. whose response to positive and negative voltage scans is symmetrical (as shown in **Supplementary Fig. 3**). Its conductance will rapidly decrease after removing the applied voltage (automatically from on-state to off-state). Therefore, it is a volatile non-polar memristive device.”

Supplementary Figure 3. Experimental bidirectional quasi-static voltage scanning of a VO₂ device (10 cycles).

2) The quasi-static IV curve has a voltage window between 0 and 2V. The devices are then used in faster dynamics. There is no characterization of the device for pulse stimuli. I suggest you to better study and present the electrical response. For many memristive technologies, as volatile Ag-based RRAMs and Ovonic-threshold switching technologies, when the dynamic is fast the voltage required for the switching increases. Did you see similar effects?

Our response: We would like to thank the reviewer for raising the constructive comments. As shown in the quasi-static IV curve, the VO₂ device will undergo the phase transition at a threshold voltage, it is necessary to study the transient process of devices to understand its switching dynamics and time constraints for computing. When we parallel connect a capacitor to the VO₂ device to form an oscillator, the oscillation frequency mainly depends on the rate of charge accumulation and release on the capacitor. However, the maximum oscillation frequency is limited by the transition speed of the device.

Therefore, we add the characterization of the pulse stimuli. The testing circuit diagram is shown in **Supplementary Figure 6a**. The voltage pulse is from the SPGU unit of the Agilent B1500A semiconductor parameter analyzer. We gradually increase the pulse voltage from the threshold voltage and recode the time difference between voltage input and current response. The testing result is shown in **Supplementary Figure 6b-h**.

The volatile Ag-based atomic-switching TS device's switching mechanism is the formation and dissolution of unstable metal filaments made injected active electrode.

The B-Te-based Ovonic threshold device's switching mechanism originated in an electronic phenomenon with a secondary thermal effect. Their delay time will decrease exponentially as the larger voltage is applied to the devices. We can observe similar characteristics in the planer VO₂ device. When the voltage increases from the threshold voltage of the device (4.5 V), the switching time will first have a rapid reduction from 300ns to 120 ns when the pulse voltage is around the V_{th}. However, when the applied voltage is greater than 4.8V, the switch speed comes to saturation (around 110 ns). The relationship between switching time and pulse voltage is shown in **Figure R6**.

Figure R6. The relationship between off-state to on-state switching time and pulse voltage.

To address this question, we have revised the following sentences into Page 12-13 and Supplementary Figure 6 of the revised manuscript:

“Furthermore, the VO₂ device's electrical response to pulse stimuli is important to dynamic oscillation. **Supplementary Fig. 6a** shows the testing circuit diagram of the devices' switching time. We gradually increase the pulse voltage from the threshold

voltage and recode the time difference between voltage input and current response. The testing result is shown in **Supplementary Fig. 6b-h**. It can be observed when the voltage increases from the threshold voltage of the device (4.5 V), the device will have a transition from off-state to on-state. The switching time will first have a rapid exponential reduction from 300ns to 120 ns when the pulse voltage is around the V_{th} . The similar effects can also be observed in other threshold devices, like Ag-based atomic-switching TS device and B-Te-based Ovonic threshold device⁴³. When the applied voltage is greater than 4.8V, the switch speed becomes saturation (around 110 ns).”

Supplementary Figure 6. Experimental measurements of VO₂ device’s switching time from off-state to on-state. (a) The testing circuit diagram (b-h) Time difference between the voltage₂ input and the current output when the VO₂ device switches from off-state to on-state, whose threshold voltage is around 4.5 V.

3) Since the hysteresis present in the IV curve, is there a retention behavior of the

switching? Which is the switching time? I suggest to characterize the switching in pulsed dynamic and better highlight the switching time and the retention properties, which limit the overall working frequency.

Our response: We would like to thank the reviewer for the constructive comments. The VO₂ device is a volatile device, whose high conductivity state cannot be retained after the applied voltage is smaller than V_{hold}. The retention time is very short (around 100 ns). We also add pulse dynamic experiments to measure the retention time because the switching dynamic limits the overall working frequency. The testing circuit diagram is shown in **Supplementary Figure 7a**. We add 10 us voltage pulse with different voltage amplitude and record the time difference between voltage input and current response when the voltage pulse is removed. The detailed result is shown in **Supplementary Figure 7b-h**.

It can be observed that the switching time is settled around 90 ns. The amplitude of the applied voltage influences less on the device's retention process. We draw the switching time of off- to on-state and on- to off-state in one figure, as shown in **Supplementary Figure 8**.

Therefore, the maximum delay time is around 390 ns. The maximum frequency limit of the VO₂ devices is theoretically 2.6 MHz. Due to the parallel connection of a sensing capacitor in our design, the single oscillator's frequency is around 120 kHz and when they are coupled the frequency is around 80 kHz.

To address this question, we have revised the following sentences into Page 13 of

the revised manuscript:

“If the voltage pulse is removed, the VO₂ device will quickly switch back from on-state to off-state. Voltage pulses with different voltage amplitudes are applied on the VO₂ device for 10 us and the time difference between voltage input and current response is recorded, as shown in **Supplementary Figure 7b-h**. It can be observed that the switching time is settled around 90 ns. The amplitude of the applied voltage influences less on the device’s retention process. We draw the switching time of off-state to on-state and on-state to off-state in **Supplementary Figure 8**. The devices’ switching time and retention properties limit the overall working frequency. As for the VO₂ device in this work, the maximum working frequency is around 2.6 MHz.”

Supplementary Figure 7. Experimental measurements of VO₂ device’s switching time from on-state to off-state. (a) The testing circuit diagram (b-h) Time difference between the voltage input and the current output when the VO₂ device switches from on-state to off-state, whose threshold voltage is around 4.5 V under pulse stimulation.

Supplementary Figure 8. Experimental measurements of the relationship between the VO₂ device's switching time and the applied pulse voltage. The switching time limits the maximum frequency (around 2.6 MHz).

4) The quasi-static IV curve shows large currents in the range of mA. How did you compute the overall pulse energy?

Our response: We would like to thank the reviewer for raising the valuable questions.

The energy consumed mainly in the fire and reset process. The oscillators' circuit is shown in **Figure R7**

Figure R7. The VO₂ oscillator's circuit.

It can be described by:

$$C_s \frac{dV_{out}}{dt} = \frac{V_{dd} - V_{out}}{R_L} - \frac{V_{out}}{R_{VO_2}}$$

When we measure the output voltage V_{out} , the total current can be computed by:

$$I(t) = (V_{dd} - V_{out}(t))/R_L$$

Then the energy consumption can be computed by:

$$P = \frac{1}{T} \times \int_0^T V_{dd} \times I(t) dt$$

Therefore, the per spike energy can be computed by: $E = P/frequency$

When we obtain the oscillation data from experiments, the energy per spike can be computed in this way. The current through the VO₂ is a sharp spike at the falling edge of voltage oscillation. When the oscillators are coupled, this method is also suitable because the coupling capacitor doesn't cost extra power in a cycle. According to our current estimation. The energy per spike of the VO₂ device oscillators can reach 2 nJ/per spike when the V_{dd} is 6 V and oscillation frequency is 1 MHz, resulting in a substantial power consumption of about 2 mW. In our gesture recognition task, the power consumption is measured at 7.56 mW for three coupled VO₂ oscillators. However, despite this relatively high power consumption, the system offers low latency, resulting in a low Energy-Delay Product (EDP) of approximately 3.07 pJs in gesture recognition tasks. The EDP is the multiply of energy and time, which is the key comparative metric

in neuromorphic computing systems.

Besides, based on the quasi-static IV curve, we can also extract the high/low resistance of the device and the threshold voltage and hold voltage. Then we can build a threshold transition device model in LTspice. The energy consumption can be obtained by IV integrating. For example, $V_{th} = 3.6V$, $V_{hold} = 1.5 V$, $R_{on}=150 \text{ ohm}$, $R_{off} = 20 \text{ kohm}$. $R_0 = 1 \text{ ohm}$, $C_0 = 5 \text{ nF}$. The R_0 and C_0 is to suppress instantaneous state transitions in LTspice simulation.

Regarding the application:

5) there are several papers which propose gesture recognition using memristive architecture, with more complex architecture and more computational power. A comparison with such works may better highlight your claim.

1. Zhong, Y., Tang, J., Li, X. et al. A memristor-based analogue reservoir computing system for real-time and power-efficient signal processing. Nat Electron 5, 672–681 (2022). <https://doi.org/10.1038/s41928-022-00838-3>

2. B. Chen, J. Yao, J. Xia, R. Yang and X. Miao, "A Strain-Sensitive Flexible MoTe₂-Based Memristor for Gesture Recognition," in IEEE Electron Device Letters, vol. 44, no. 4, pp. 622-625, April 2023, doi: 10.1109/LED.2023.3249340.

3. Ceolini, E. et al. "Hand-Gesture Recognition Based on EMG and Event-Based Camera Sensor Fusion: A Benchmark in Neuromorphic Computing". Frontiers in Neuroscience vol. 14 (2020). <https://doi.org/10.3389/fnins.2020.00637>

Our response: We would like to thank the reviewer for the constructive comments.

The reference 1 implements a fully analogue reservoir computing system that composes the dynamic memristors as the first layer and the non-volatile memristor cross as the second layer. 24 memristors are used as the first layer reservoir. The total energy is 22.2 uW of the entire processing system and the delay time is 1.6 ms for the dynamic memristor. However, it doesn't consider the sensing signal acquisition. Here, in this work, we can only use 3 coupled VO₂ devices to encode the sensing signals into phase patterns for sensory pre-processing. The energy is 7.68 mW. The delay time of the transient process from the initial state to the stable coupling state is only 20 μs.

The reference 2 design non-volatile MoTe₂ memristors, which can combine sensor function and computing function in a single device. An artificial neural network is applied to do gesture recognition, which needs hundreds of devices($14*25+25*6 = 500$).

There is no power estimation because the application is based on simulation due to the large number of needed devices.

The reference 3 design a Hand-Gesture Recognition of EMG signals based on CMOS neuromorphic hardware. The dynamic power is 29.4 mW and the processing time is 5.89 ms. This paper uses the EDP (Energy Delay Product) as the key comparative indicators in neuromorphic computing systems.

We have included a detailed comparison of our approach to gesture classification with methods based on memristors and CMOS in **Supplementary Table 2**. This comparison highlights three key advantages:

- 1. Mitigating Device Variation:** The proposed computing approach utilizing coupled oscillation effectively addresses the issue of large device-to-device variation. Differences in the oscillators' natural frequencies can be encoded as a single coupled frequency with varying phases, and the coupling capacitor is the key factor that determines the phase difference. In general, variations in device characteristics primarily result in fluctuations in oscillation frequencies. Nevertheless, within our specific application, the phase difference of coupled oscillations assumes a pivotal role. Fluctuations in the oscillation frequencies of distinct devices will be mitigated through capacitive coupling since they will ultimately synchronize to the same frequency.
- 2. Power Efficiency:** While VO₂ oscillators have relatively high power consumption due to their high threshold voltage and low on-state resistance, we acknowledge that there is room for improvement in this regard. Furthermore, we have highlighted the short delay time in coupled oscillation, defined as the time for the transient process from the initial state to the stable coupling state, which is only 20 μs in our work (as demonstrated in **Supplementary Figure 17**).
- 3. Scalability:** Crucially, our approach can scale effectively to larger systems. N coupled VO₂ oscillators can encode 2ⁿ distinguishable modes, offering a significant advantage over other dynamic processing methods that require exponential device consumption as system size grows.

To address this question, we have revised the **Supplementary Table 2** and the **sentences in page 25**:

“Couple VO₂ oscillators can have stronger computing superiority compared to the other methods including non-volatile memristors and CMOS circuits. A detailed comparison table with previous works is shown in Supplementary Table 2. The volatile memristive devices including VO₂, NbO₂ suffer from the large device-to-device variation. Different VO₂ devices exhibit notable device-to-device variation even at the same size, which can be disadvantageous when used as individual oscillators. However, the coupling oscillation strategy we adopt enables the synchronization of two oscillators with distinct intrinsic oscillation frequencies to a common frequency. Subsequently, we encode and classify them based on their phase patterns. The alteration in sensing capacitance exerts a more pronounced impact on the phase difference after the oscillators are coupled. Consequently, this strategy serves to the mitigation of device-to-device variation. Besides, due to the high threshold voltage and low on-state resistor, the VO₂ oscillators have a high power, which can be improved in the future. The delay time in coupled oscillation can be defined as the time of the transient process from the initial state to the stable coupling state, which is only 20 μ s in this work (as shown in Supplementary Fig. 3). Most importantly, when expanding to larger systems, n coupled VO₂ oscillators can encode 2^n distinguishable modes while the other dynamic processing methods require exponential device consumption.”

Supplementary Table 2. A comparison table of gesture recognition task with previous works including memristors and CMOS.

Zhong, Y. et al	Chen, B., et al	Lu L. et al	Ceolini, E. et al.	Rui, Y. et al	This work
--------------------	-------------	-----------------------	---------------	------------------

Device	TiO _x	MoTe ₂	P3HT/PEO	CMOS	VO ₂	VO ₂
Computing style	Near-sensor computing	In-sensor computing	In-sensor computing	Near-sensor computing	Near-sensor computing	Near-sensor computing
Computing method	Reservoir computing	Artificial neural network	Artificial neural network	Spiking neural network	Spiking neural network	Coupled oscillation
Memristor type	Volatile	Non-volatile	Non-volatile	/	Volatile	Volatile
D2D variation aware	NO	NO	NO	/	NO	YES
Device number	24M+2048M(experiment)	500M (simulation)	5(experiment)*512*128*84*3(simulation)	/	10R+5C+5M(experiment)	3R+3M+3C(experiment) 3 (simulation)
Power	22.2 uW	/	/	29.4 mW	/	7.68 mW
Time	1.6 ms	/	/	5.89 ms	/	20 us
Energy	13.32 nJ	/	/	173.2 uJ	/	153.6 nJ
EDP*	21.31 pJ*s	/	/	1.02 uJ*s	/	3.07 pJ*s

*The EDP (Energy Delay Product) is the multiply of energy and time, which is the key comparative indicators in neuromorphic computing systems.

6) The decision making module, as well as the force learning, should be explained more in detail, to provide a better comparison for future work and comparison. Which kind of architecture are? How many layers or building-blocks? Are there some references? Supplementary figure 7 is not explained, and the variable inside neither. In the energy consideration it should be included the also the second layer (force learning or decision making).

Our response: We would like to thank the reviewer for the constructive comments.

The decision-making module consists of connected dynamic neurons with self-feedback, as shown in **Fig. 1d**. When a neuron outputs maximum firing strength, one mode of input can be easily distinguished.

There are two layers for post-processing, one is a linear combination and the other is a decision-making module. We provide the mathematical link between the detail as shown **Fig. 3**.

We mainly build the decision-making module based on the two references (*J. Neurosci.*, **26**, 1314-1328 (2006) & *Neural Netw.*, **143**, 74-87 (2021)). We add a more detailed explanation about the decision-making module and the mathematical links between the coupled oscillators and the decision-making module in the revised manuscript.

We add the explanation in Supplementary Fig. 15. It shows the detailed algorithm flow in post-processing.

In the energy consideration, we only consider the coupled VO₂ oscillators because they represent the process of near-sensor computing and the decision-making is a post-processing procedure. According to their nonlinear mathematical equations, it's hard to evaluate a reliable power consumption.

To address this question, we have revised the sentences in page 8-9 and Fig. 1d, Fig. 3a, Supplementary Fig. 15:

The decision-making network we employed is a simplified mean-field decision-making model³⁴⁻³⁵. In the decision-making network illustrated in **Fig. 1c**, there are n neurons each representing one of the classification decisions of the sensory pattern. Among the neurons, mutual inhibition is introduced to make them compete with each other and determine the only winner as the final result. The model can be mathematically described by the following equations³⁵:

$$x_i(t) = J_E s_i + \sum_{j \neq i}^{N_{dm}} J_M s_j + I_i \quad (1)$$

$$r_i = \frac{\beta}{\gamma} \ln \left[1 + e^{\frac{x_i - \theta}{\alpha}} \right] \quad (2)$$

$$\tau_s \frac{ds_i}{dt} = -s_i + \gamma(1 - s_i)r_i \quad (3)$$

where x_i and r_i denotes the synaptic input and neuronal activity of the i th neuron. s_i represents the synaptic. The equation (1) describes the neurons in the decision-making module that receive synaptic current from themselves feedback $J_E s_i$, other neuron $\sum_{j \neq i}^{N_{dm}} J_M s_j$ and the input from the former layer Z_i . The equation (2) describes the neuron's nonlinear activation function with a threshold θ . The equation (3) describes the slow dynamics of the synaptic current. Specifically, the synaptic input x_i described by Equation 1 is composed of three parts, which are the self-excitation $J_E s_i$ ($J_E \geq 0$), total recurrent input from other neurons with $J_M \leq 0$ indicating mutual inhibition and the feedforward input I_i from the ONN module. The neuronal activity r_i can be further calculated by x_i using the nonlinear activation function in Equation 2 with parameters α , β , γ , and threshold θ . The slow dynamics of the synaptic current originating from the activity-dependent receptors is formulated in Equation 3, which plays a crucial role in the spatiotemporal information processing of the decision-making network.

The time constant τ_s ($\gg 1$) in Equation 3 controls the time window for integrating input over time by the decision-making neurons. The first network's output is the voltage oscillation signal V , which is the $((O1(t), O2(t), \dots, On(t)))$ in **Fig. 1c-d**. First, the signal V through linear combination converts to Z . The impact of the linear combination is to adjust the signal to an appropriate range. The weight of the linear combination can be trained by Force learning³⁵. Then Z is the input to the decision-making module as shown in Equation 1. The expected output \mathbf{r} of the decision-making module is that one neuron fires with maximum strength while the others fire less.

Fig. 1 The sensory processing framework in different systems. a Biological sensory system. Receptors and neurons convert physical stimuli into electrical pulses which are transmitted to the cortex for high-level processing. **b** Conventional sensory system. In digital processing way, the signals from sensors are first digitalized through ADC and then put into separate memory units and processing units. In the ANN way, a crossbar composed of sensors can perform linear operations and realize classification. **c**

Dynamic sensory system. The proposed sensory system consists of a memristive oscillatory network and a decision-making module, converting discrete oscillator frequencies into phase differences at the same frequency through coupling. **d** Hardware mapping of the dynamic sensory system. The specific circuit diagram can encode the sensory information in the phase pattern after electrical coupling.

Fig. 3 Touch recognition with the bio-inspired sensory recognition system. a Flow chart of the coupled VO₂ memristive oscillators (in experiment), the decision-making module(in software), and the mathematical link between them. **b** The phase pattern without touch event. The outputs of the two neurons in ONN are not synchronized to the identical frequency. **c** The phase pattern with touch event. The outputs of the two neurons in ONN are locked in frequency with stable difference in phase. **d** Outputs of the decision-making network corresponding to the non-synchronized condition

(without touch). e Outputs of the decision-making network corresponding to the synchronization pattern (with touch).

Supplementary Figure 15. Detailed decision-making module and FORCE learning algorithm processing flow in high-level processing to post-processing continuous time signals from ONN. In the inference process, the voltage oscillation signals X from the coupled oscillators' first pass through the linear combination and turn to Z . The signal Z is the input of the decision-making module, computing the synaptic input and computing a nonlinear activation function, turning to r as the output in the end. Because all the signals are time-continuous, the slow volatile dynamic of synapses are also included to compute s , which can be seen as the synapse current in inter-connection and self-connection. In the Force Learning training process, we set training target output Z , and compute the error e between target Z and actual output C . Then using the error function to adjust the weight of the linear combination W with an update matrix P .

7) The application is based on 3 coupled neurons. What happens for larger system?

Why didn't you exploit the full set of 8 different gestures you can encode (as in

Supplementary figure 9)? Is it possible to use a simple synaptic layer (made by other memristive devices) to distinguish the different gestures?

Our response: We would like to thank the reviewer for raising the valuable questions.

In this work, we experimentally demonstrate the three coupling VO₂ neurons for dynamic sensory processing. Three coupling VO₂ neurons can encode 8 patterns. When the number of neurons increases to n , the total encodable mode will rise to 2^n . For example, ten neurons can encode 1024 sensory modes in phase difference. Besides, the VO₂ oscillators are nonlinear oscillators. When the scale increases, the population can generate new emergence phenomena that need to apply complex system theory. Combinatorial optimization problems, such as Ising machine models, can also be mapped on the coupled nonlinear oscillators. In this work, we experimentally demonstrate the three coupled VO₂ oscillators, and the eight encoding modes are shown in **Fig.5**. Due to coupling characteristics, the output voltage signals can form clear trajectories in phase space, which reflect the classifiable modes as shown in **Fig.5 a, c, e, g, i, k, m, o**. The trajectories and their projections on x-y, y-z, x-z planes present the dynamic of stable limit cycles, whose shapes reflect the phase differences of different oscillators. The eight kinds of limit cycles attach great importance to dynamic computing based on memristors. It shows the actual physical memristors with variations can be accurately controlled in population dynamics.

To prove that the eight feature modes can be classified in the proposed dynamic systems, we expand the scale of the decision-making module to 8 neurons, and each neuron

presents one mode. The software simulation is shown in **Fig.5 b, d, f, h, j, l, n, p**. When voltage signals in one mode are linear weighted (3×8) and input into the module, the neuron corresponding to this combination will fire with maximum and suppress the other neurons. The results show the decision-making module can handle such dynamic classification problems while other methods like recurrent neural networks need hundreds of neurons in hidden layers.

A simple synaptic layer is not enough to distinguish the different gestures because the problem is time-continuous. If using a Recurrent Neural Network, the network will need a large number of hidden neurons. Reservoir computing can be used in such problems. However, as shown in the Supplementary Table 2, the delay time of reservoir computing is relatively high, and our coupled oscillation approach can provide better EDP (Energy Delay Product).

To address this question, we have added the **Fig. 5** and revised the following sentences on Page 13 of the revised manuscript:

“In this work, we experimentally demonstrate the three coupled VO_2 oscillators, and the eight encoding modes are shown in **Fig.5**. Due to coupling characteristics, the output voltage signals can form clear trajectories in phase space, which reflect the classifiable modes as shown in **Fig.5 a, c, e, g, i, k, m, o**. The trajectories and their projections on x-y, y-z, x-z planes present the dynamic of stable limit cycles, whose shapes reflect the phase differences of different oscillators. The eight kinds of limit cycles attach great importance to dynamic computing based on memristors. It shows the actual physical memristors with variations can be accurately controlled in

population dynamics. To prove that the eight feature modes can be classified in the proposed dynamic systems, we expand the scale of the decision-making module to 8 neurons, and each neuron presents one mode. The software simulation is shown in **Fig.5 b, d, f, h, j, l, n, p**. When voltage signals in one mode are linear weighted (3×8) and input into the module, the neuron corresponding to this combination will fire with maximum and suppress the other neurons. The results show the decision-making module can handle such dynamic classification problems while other methods like recurrent neural networks need hundreds of neurons in hidden layers.”

Fig. 5 The trajectories in the phase plane of three coupled VO₂ oscillators in eight coupled oscillation modes and the decision-making classification results. a

The experimental trajectory in the phase plane of case '000'. **b** The decision-making classification results in the software of case '000'. Besides, the experimental trajectories in the phase plane of case '100', '010', '001', '110', '011', '101', '111' correspond to **c, e, g, i, k, m, o**. The decision-making classification results in software of case '100', '010', '001', '110', '011', '101', '111' correspond to **d, f, h, j, l, n, p**. Three coupled VO₂ dynamic devices can encode eight classifiable sensory modes in phase space.

8) How many attempts did you perform for each gesture? Did you consider also some cycle-to-cycle and device-to-device variability?

Our response: We would like to thank the reviewer for raising the valuable questions.

In the experimental demonstration, we performed two or three times for each gesture. Because the data is time continuous, we have enough data for Force Learning training and do classification further.

The high cycle-to-cycle uniformity is necessary for coupled oscillators. The practical experimental demonstration needs a highly uniform oscillation frequency so that the information can be permanently encoded in phase differences. The cycle-to-cycle variation of VO₂ device is shown in **Supplementary Figure 4**. The deviation of the threshold voltage is 0.00021 and the deviation of the hold voltage is 0.00013. The reason we adopt VO₂ oscillators is their high cycle-to-cycle uniformity. For practical experimental demonstrations, it is crucial to achieve a highly uniform oscillation

frequency to enable the reproducible encoding of information through phase differences and consistently output a constant frequency.

Our coupled oscillation approach can suppress the device-to-device variation. The volatile memristive devices including VO₂, NbO₂ suffer from the large device-to-device variation. We characterized 40 VO₂ devices to study their device-to-device variation. The threshold voltage under quasi-static voltage sweep varies from 3.55 V to 4.10 V and the hold voltage varies from 1.52 V to 1.96V. Importantly, our proposed computing approach involving coupled oscillations effectively mitigates this device-to-device variation. This is achieved by encoding the differences in the oscillators' natural frequencies as a single coupled frequency with varying phases. Subsequently, we encode and classify them based on their phase patterns. The alteration in sensing capacitance exerts a more pronounced impact on the phase difference after the oscillators are coupled. Consequently, this strategy serves to the mitigation of device-to-device variation.

In general, variations in device characteristics primarily result in fluctuations in oscillation frequencies. Nevertheless, within our specific application, the phase difference of coupled oscillations assumes a pivotal role. Fluctuations in the oscillation frequencies of distinct devices will be mitigated through capacitive coupling since they will ultimately synchronize to the same frequency

To address this question, we have added **Supplementary Figure 4-5** and revised the following sentences into Page 12 and Page 27:

“The monocrystalline VO₂ memristor provides high cycle-to-cycle variation

(Supplementary Fig. 4). The deviation of the threshold voltage is 0.00021 and the deviation of the hold voltage is 0.00013. The high cycle-to-cycle variation is necessary for the practical oscillation demonstration because the oscillator needs to output a single constant frequency.”

Supplementary Figure 4. Measurement of VO₂ device cycle-to-cycle variation. (a) quasi-static voltage scanning of a VO₂ device (200 cycles). (b) The cycle-to-cycle variation of VO₂ device’s threshold voltage and hold voltage.

“40 VO₂ devices are characterized to study their device-to-device variation, which will affect different natural frequencies of VO₂ oscillators, as shown in **Supplementary Fig. 5**. The threshold voltage under quasi-static voltage sweep varies from 3.55 V to 4.10 V and the hold voltage varies from 1.52 V to 1.96V. Devices with similar performance are more likely to form out-of-phase synchronization after capacitive coupling.”

Page 27,

“The volatile memristive devices including VO₂, NbO₂ suffer from the large device-to-device variation. Different VO₂ devices exhibit notable device-to-device variation

even at the same size, which can be disadvantageous when used as individual oscillators. However, the coupling oscillation strategy we adopt enables the synchronization of two oscillators with distinct intrinsic oscillation frequencies to a common frequency. Subsequently, we encode and classify them based on their phase patterns. The alteration in sensing capacitance exerts a more pronounced impact on the phase difference after the oscillators are coupled. Consequently, this strategy serves to the mitigation of device-to-device variation.”

Supplementary Figure 5. Measurement of VO₂ device device-to-device variation in 40 devices.

9) Supplementary figure 6 is not cited in the main text

Our response: We would like to thank the reviewer for the comments.

Supplementary Figure 14 shows the synchronization dynamic between three

memristive sensory neurons through capacitances under different oscillators natural frequencies. We keep two oscillators' natural frequency same and change the other oscillator's nature frequency by changing the series resistor R_3 .

The $R_1=R_2=10.2$ k Ω . When changing R_3 , the phase differences present monotonic changes when $R_3 < 10.8$ k Ω . Especially, when $R_3 = R_1=R_2=10.2$ k Ω , the three coupled oscillation frequencies exhibit a phase difference of $\frac{\pi}{3}$ and $\frac{2\pi}{3}$. It also shows the VO₂ device-to-device variation can be suppressed by the coupling capacitors.

To address this question, we have revised the following sentences into Page 12:

“When expanding the oscillators from two to three, **Supplementary Figure 14** shows the synchronization dynamic between three memristive sensory neurons through capacitances under different oscillators natural frequencies. We keep two oscillators' natural frequency same and change the other oscillator's nature frequency by changing the series resistor R_3 . The $R_1=R_2=10.2$ k Ω . When changing R_3 , the phase differences present monotonic changes when $R_3 < 10.8$ k Ω . Especially, when $R_3 = R_1=R_2=10.2$ k Ω , the three coupled oscillation frequencies exhibit a phase difference of $\frac{\pi}{3}$ and $\frac{2\pi}{3}$. It shows the VO₂ device-to-device variation can be suppressed by the coupling capacitors.”

REVIEWER COMMENTS

Reviewer #1 (Remarks to the Author):

The authors have made significant improvements to the manuscript and also the to the supplementary sections.

There are still some concerns about the computational theory part which are not clear and maybe authors can provide these details as either an appendix or supplementary sections. Here are the concerns:

1 - authors claim to process spatio-temporal patterns with capacitively coupled VO₂s and capacitive sensors. The sensor data is a variable capacitance. While this is interesting to see how coupled VO₂ synchronise, the frequency variations due to different sensors still remains unclear. For example, if the frequency difference is so different, oscillators tend to not lock and synchronise. Thus, how can one guarantee that different spatio-temporal signals allow for VO₂ oscillator to lock and synchronise. To allow some locking would require also changing the coupling capacitance to adapt to the external signals. Thus, only limited capacitive sensors values would work and it is rather unclear what would happen if natural frequencies and capacitive frequencies are order of magnitude different. Some analysis on this would reveal under what conditions the proposed approach would work.

2 - device to device and cycle to cycle variability are two of the main challenges in locking VO₂ oscillators. Yet this is not clear how these variabilities will impact the sensor processing. How would these variabilities also impact the resolution of phases that can be measured and ultimately how much it can sense (capacitive ranges for sensing)

3 - how would the circuit scale up beyond 3 oscillators. A general formulation (analytical) would reveal what can be expected from a system of n coupled oscillators in ideal form. Providing a general analytical formulation would offer insights into the expected behavior of a system comprising n coupled oscillators in its ideal state.

4 - how would one determine the coupling capacitive values for given capacitive sensors

5- experimentally, have different frequency and capacitive ranges were applied and synchronisation issues observed

6 - computation of energy consumption should be derived in the appendix to show clearly the energy efficiency gain as it is written is over 1000x better than CMOS counterpart. Please show this in detail and also cite relevant references.

These clarifications and analyses would significantly enhance the comprehensibility and robustness of the manuscript.

Reviewer #2 (Remarks to the Author):

Authors revised the abstract and it is in acceptable form now. However, overall, the paper is not well-written. I had a comment on wrong terminology of calling neurons memristive and the authors did not understand my comment and instead wrote a long response on history of memristors which was totally unnecessary as what memristor is well-known to all the reviewers. I think this is just an oscillatory neuron, so terminology needs to be corrected. Fig. R1 suggests there were issues with the fabrication. Experiment setup is a breadboard (Sup Fig 11). Doesn't it suffer from parasitics? It is hard to understand why the authors could not design a PCB to integrate everything on it.

Reviewer #3 (Remarks to the Author):

Thanks Authors for the the great job and the efforts you did for the revision part. The comments were well addressed and the paper is now more clear and better explained.

MS No: NCOMMS-23-18953A

Title: High-order sensory processing nanocircuit based on coupled VO₂ oscillators

Response to Reviewer's Comments

Reviewers' comments are in black color

Responses are in blue color.

Changes made in main text and SI are in green color

Reviewer #1 (Remarks to the Author):

The authors have made significant improvements to the manuscript and also to the supplementary sections.

There are still some concerns about the computational theory part which are not clear and maybe authors can provide these details as either an appendix or supplementary sections.

Here are the concerns:

Our response:

We appreciate the valuable feedback provided by the reviewer. We have carefully considered the concerns regarding the computational theory part, and we agree that providing additional details in the supplementary sections would enhance the clarity of the manuscript. In response, we have enhanced the theoretical section and revised the manuscript according to all reviewer suggestions, as detailed below.

1 - authors claim to process spatio-temporal patterns with capacitively coupled VO₂s and capacitive sensors. The sensor data is a variable capacitance. While this is interesting to see how coupled VO₂ synchronise, the frequency variations due to different sensors still remains unclear. For example, if the frequency difference is so different, oscillators tend to not lock and synchronise. Thus, how can one guarantee that different spatio-temporal signals allow for VO₂ oscillator to lock and synchronise. To allow some locking would require also changing the coupling capacitance to adapt to the external signals. Thus, only limited capacitive sensors values would work and it is rather unclear what would happen if natural frequencies and capacitive frequencies are order of magnitude different. Some analysis on this would reveal

under what conditions the proposed approach would work.

Our response:

We would like to thank the reviewer for the insightful comments and raising important concerns regarding the potential challenges associated with frequency variations due to different sensors in our proposed approach. We appreciate the opportunity to address these concerns and provide further clarification.

The issue of frequency differences among sensors is indeed a critical aspect of oscillator synchronization. We acknowledge that significant frequency disparities can hinder the locking and synchronization process. To address this concern, we have conducted a detailed analysis to investigate the conditions under which our proposed approach can effectively achieve synchronization despite variations in natural frequencies and capacitive frequencies.

Firstly, when the disparity between distinct eigenfrequencies is too significant, they will fail to couple. In the experiment, this is evidenced by the sensing capacitance value connected in parallel with the device. If the discrepancy in capacitance values is considerable, diverse oscillators will be unable to establish synchronization.

In **Supplementary Note 2**, we introduced a theoretical model on coupled oscillations and extended our exploration to a broader parameter range. Through simulation, we initially examined the extent of the difference in sensing capacitance values that would lead to the loss of coupling when the coupling capacitance is fixed. As shown in Supplementary Fig 21 and Supplementary Fig 23. As for the device model, $V_{th} = 3.8$ V, $V_{hold} = 1.6$ V, $R_{off} = 20$ kohm, $R_{on} = 300$ ohm.

For two-oscillator coupling:

We set $V_{dd} = 8$ V, $R_L = 10$ kohm, $C_o = 300$ pF. We set one sensing capacitor is 400 pF, so that the natural frequency of a single oscillator is 402 kHz. Then we change the other oscillator's sensing capacitor to observe the two oscillators' coupling situation. The result is shown in Supplementary Fig 21. Therefore, only when the other oscillator sensing capacitor is in 310 pF- 500 pF, the two oscillators can be synchronized and lock phase difference. The difference in natural frequency is 117 kHz (310pF) and 80kHz (500pF) in this situation.

Supplementary Figure 21. Influence of sensing capacitor to two oscillators' coupling. The green background indicates synchronization, while the gray background suggests desynchronization.

When increasing the coupling capacitance, the tolerance of the natural frequency difference will also increase slightly. As can be seen in Supplementary Fig 22 (a), when the coupling capacitance is increased to 500 pF, the natural frequency range is from 310 kHz (510 pF) to 536 kHz (300 pF); when the coupling capacitance is increased to 1 nF, the natural frequency range is from 293 kHz (500 pF) to 555 kHz (290 pF). However, due to the real physical devices having device-to-device variation, the coupling capacitance cannot be too large, otherwise the synchronization will be destroyed. As shown in Supplementary Fig 22 (b), when device 1 and device 2 have different V_{th} , with $V_{th1} = 3.8$ V, $V_{th2} = 3.7$ V, and the coupling capacitance increases to 2 nF, the two devices cannot synchronize.

Supplementary Figure 22. Influence of coupling capacitor to two oscillators' coupling without (a) and with (b) device to device variation.

For three-oscillator coupling:

The parasitic capacitance is set at 200 pF. As shown in Supplementary Fig 23, With the two devices' parallel capacitors fixed at 0.4nF, varying the third capacitance value revealed a valid coupling range in 0.3nF-0.7nF.

Supplementary Figure 23. Influence of sensing capacitor to three oscillators' coupling.

Furthermore, when two devices' parallel capacitors were set at 0.4nF and one device's parallel capacitor was set at 0.6nF, we modified the coupling capacitor value to observe potential

coupling. As illustrated in Supplementary Fig 24 (a) and (b), we observed that coupling persisted until the value increased to 700 pF and 800 pF in case of mode 001 and 011 respectively.

Supplementary Figure 24. Influence of coupling capacitor to three oscillators' coupling in case of mode 001(a) and 011(b).

Overall, we performed simulations to validate and ensure the practical feasibility of the proposed approach under various conditions. By conducting a comprehensive analysis and providing empirical evidence, we conducted the robustness and limitations analysis of our method in the presence of diverse capacitive coupling and varying sensor characteristics. Again, we appreciate the reviewer's constructive feedback, and we have been committed to addressing these concerns through rigorous analysis and simulation, we hope reviewers will find this supplementary content satisfactory.

2 - Device-to-device and cycle-to-cycle variability are two of the main challenges in locking VO₂ oscillators. Yet this is not clear how these variabilities will impact the sensor processing. How would these variabilities also impact the resolution of phases that can be measured and ultimately how much it can sense (capacitive ranges for sensing)

Our response:

We would like to thank the reviewer for highlighting the critical issues of device-to-device and cycle-to-cycle variability in VO₂ oscillators and their potential impact on sensor processing.

In our revised manuscript, we have conducted detailed analysis to assess the impact of these variabilities on the resolution of phase measurements and the overall sensing performance.

Specifically, we have investigated how variations between different devices and cycles may affect the precision and accuracy of phase measurements. Device-to-device variation does indeed impact coupling, and devices with excessively large differences may fail to establish synchronize. The primary manifestation of device-to-device variation lies in V_{th} . In our supplementary simulation, as shown in **Supplementary Fig 25**, we configured the coupling state to 001 mode, setting the V_{th} of two devices simultaneously to 3.8 and manipulating the V_{th} of the third device. The result shows that coupling is unattainable when V_{th} is below 3.5, but it remains feasible when V_{th} exceeds 4.1V. This outcome substantiates that coupling is established within the range of device fluctuations ($3.5 < V_{th} < 4.1V$). Note that the sudden shifts in the frequency of synchronous coupling in case of mode 000 and 011 imply additional dynamic features that extend beyond the scope of this manuscript and will be elaborated in our subsequent work.

Supplementary Figure 25. Influence of device-to-device variation of V_{th} on three oscillators' coupling in case of mode 000(a), 001(b), 011(c) and 111(d).

Cycle-to-cycle variation will influence the phase resolution post coupling. The phase jitter

induced by V_{th} fluctuate during the oscillation represents the lower limit of phase resolution. As shown in Supplementary Fig 26, we have evaluated the impact of cycle-to-cycle variation in different modes. When the cycle-to-cycle variation increases from 0.01 to 0.11, the maximum value of the phase difference is less than 0.35° . Considering that the cycle-to-cycle variation of the device is within 0.01 (Supplementary Fig 4), The impact of this part is almost negligible.

Supplementary Figure 26. Influence of cycle-to-cycle variation of V_{th} on three oscillators' coupling in case of mode 000, 001, 011 and 111.

This analysis involved simulation studies quantify the extent to which these variabilities can be mitigated or compensated within the proposed framework.

3 - how would the circuit scale up beyond 3 oscillators. A general formulation (analytical) would reveal what can be expected from a system of n coupled oscillators in ideal form. Providing a general analytical formulation would offer insights into the expected behavior of a system comprising n coupled oscillators in its ideal state.

Our response:

We would like to thank the reviewer for insightful suggestion regarding the scalability of our proposed circuit beyond three oscillators. Understanding the behavior of a system with n coupled oscillators in an idealized form is crucial for assessing the practicality and limitations

of the proposed approach.

In response to this, we developed a general analytical formulation that captures the expected behavior of a system comprising n coupled oscillators. This formulation provides insights into the dynamics of the system, offering a theoretical foundation for scaling up the circuit.

$$C_{ii} \frac{dV_i}{dt} = \frac{V_{dd} - V_i}{R_0} - \frac{V_i}{R_{VO_2i}} + \sum_{j \neq i} C_{ij} \frac{d(V_j - V_i)}{dt}$$

The V_i is the i -th oscillator's output. The R_0 is the series resistor. The R_{VO_2i} is the resistor of VO_2 device of i -th oscillator. The C_{ii} is the parallel sensing capacitor of i -th oscillator and the C_{ij} is the coupled capacitor between i -th oscillator and j -th oscillator. The theoretical framework is detailed in **Supplementary Note 2**.

With this general analytical framework, we can identify trends, dependencies, and potential challenges that may arise as the number of oscillators increases. We believe this analysis would contribute to a deeper understanding of the scalability of our proposed circuit, guiding future implementations and offering valuable insights for experimental design.

4 - how would one determine the coupling capacitive values for given capacitive sensors

Our response:

We would like to thank the reviewer for raising this constructive question. In the supplementary material section of the revised manuscript, we present a model for the systematic analysis of the coupling of n oscillators. This model takes into account the arbitrary connection of n oscillators and enables the analysis of device fluctuations and nonlinearity. We are confident that this model enhances the comprehension and analysis of coupled oscillation behaviors. Through this model, we can precisely determine the coupling capacitance needed for coupling oscillators with varying parameters. Consider the coupling of two oscillators and three oscillators designed in our experiment as examples.

For two-oscillator coupling:

As shown in Supplementary Fig 22(a), when the coupling capacitance is increased to 500 pF, the natural frequency range is from 310 kHz (510 pF) to 536 kHz (300 pF); when the coupling capacitance is increased to 1 nF, the natural frequency range is from 293 kHz (500 pF) to 555 kHz (290 pF). However, due to the real physical devices having device-to-device variation, the coupling capacitance cannot be too large, otherwise the synchronization will be

destroyed. For example, when device 1 and device 2 have different V_{th} , $V_{th1} = 3.8V$, $V_{th2} = 3.7V$, and the coupling capacitance exceeds 1.6 nF, the two devices cannot synchronize as shown in Supplementary Fig 22(b).

For three-oscillator coupling:

When two devices' parallel capacitors were set at 0.4nF and one device's parallel capacitor was set at 0.6nF, we modified the coupling capacitor value to observe potential coupling. As illustrated in Supplementary Fig 24 (a) and (b), we observed that coupling persisted until the value increased to 700 pF and 800 pF in case of mode 001 and 011 respectively.

5- experimentally, have different frequency and capacitive ranges were applied and synchronisation issues observed

Our response:

We would like to thank the reviewer for raising this constructive question.

For two coupled oscillators, we experimentally verified that when the oscillators have a certain frequency difference, they fail to couple. This binary outcome serves as a means to discern the presence or absence of a pressing signal. As shown in Fig.3 of the main text in the manuscript, in the experiment here we measured two different frequencies whether the pressing signal exists or not.

For three coupled oscillators, we experimentally verified the coupling situation at a series of different frequencies. In the experiment, we controlled the two oscillators to have the same frequency, and changed the frequency of the third oscillator by adjusting the series resistance, thus observing Changes in phase difference between three coupled oscillators. The results are shown in Supplementary Fig 14.

6 - computation of energy consumption should be derived in the appendix to show clearly the energy efficiency gain as it is written is over 1000x better than CMOS counterpart. Please show this in detail and also cite relevant references.

Our response:

We would like to thank the reviewer for valuable suggestion regarding the computation of energy consumption.

The energy consumed mainly in the fire and reset process. The oscillators' circuit is shown in **Supplementary Fig 27**

Supplementary Fig 27. The VO₂ oscillator's circuit.

It can be described by:

$$C_s \frac{dV_{out}}{dt} = \frac{V_{dd} - V_{out}}{R_L} - \frac{V_{out}}{R_{VO_2}}$$

When we measure the output voltage V_{out} , the total current can be computed by:

$$I(t) = (V_{dd} - V_{out}(t))/R_L$$

Then the energy consumption can be computed by:

$$P = \frac{1}{T} \times \int_0^T V_{dd} \times I(t) dt$$

Therefore, the per spike energy can be computed by: $E = P/\text{frequency}$

When we obtain the oscillation data from experiments, the energy per spike can be computed in this way. The current through the VO₂ is a sharp spike at the falling edge of voltage oscillation. When the oscillators are coupled, this method is also suitable because the coupling capacitor doesn't cost extra power in a cycle. According to our current estimation. The energy per spike of the VO₂ device oscillators can reach 2 nJ/per spike when the V_{dd} is 6 V and oscillation frequency is 1 MHz, resulting in a substantial power consumption of about 2 mW. In our gesture recognition task, the power consumption is measured at 7.56 mW for three coupled VO₂ oscillators. However, despite this relatively high power consumption, the system offers low latency, resulting in a low Energy-Delay Product (EDP) of approximately 3.07 pJs in gesture recognition tasks. The EDP is the multiply of energy and time, which is the key comparative metric in neuromorphic computing systems. The EDP of CMOS counterpart is according to the reference [50] (Ceolini, E., et al. Hand-gesture recognition based on EMG and event-based camera sensor fusion: A benchmark in neuromorphic computing. *Front. in Neuro*, 14, 637 (2020).), which is a benchmark for hand-gesture recognition in neuromorphic computing. It shows the CMOS has the EDP of 1 uJs.

These clarifications and analyses would significantly enhance the comprehensibility and robustness of the manuscript.

To address these questions, we have revised the following sentences into **Supplementary Note 2** of the revised manuscript:

Supplementary Note 2: Simulation of the coupled oscillation

To enhance the clarity of the manuscript, especially the computational theory part. Here we supplement the theory and simulation of n coupled oscillators. Assuming that there are n oscillators coupled by capacitance. First, we can extend the formulations presented in Equations 7-9 in the manuscript to encompass more general analytical formulas, as shown below:

$$C_{ii} \frac{dV_i}{dt} = \frac{V_{dd} - V_i}{R_0} - \frac{V_i}{R_{VO_2i}} + \sum_{j \neq i} C_{ij} \frac{d(V_j - V_i)}{dt}$$

The V_i is the i -th oscillator's output. The R_0 is the series resistor. The R_{VO_2i} is the resistor of VO₂ device of i -th oscillator. The C_{ii} is the parallel sensing capacitor of i -th oscillator and the C_{ij} is the coupled capacitor between i -th oscillator and j -th oscillator.

Note that VO₂ device is a volatile memristor which has four parameters: R_{on} , R_{off} , V_{th} and V_{hold} .

$$\text{if } V_i > V_{th} \&\& R_{VO_2i} = R_{off}, R_{VO_2i} \rightarrow R_{on}$$

$$\text{if } V_i < V_{hold} \&\& R_{VO_2i} = R_{on}, R_{VO_2i} \rightarrow R_{off}$$

Through transformation, the formula can be written as:

$$\sum_j C_{ij} \frac{dV_i}{dt} - \sum_{j \neq i} C_{ij} \frac{dV_j}{dt} = \frac{V_{dd} - V_i}{R_0} - \frac{V_i}{R_{VO_2i}}$$

Using the three coupled oscillators discussed in the article as an illustration, when n=3, the formula can be simplified as:

$$\begin{pmatrix} C_{11} + C_{12} + C_{13} & -C_{12} & -C_{13} \\ -C_{21} & C_{21} + C_{22} + C_{23} & -C_{23} \\ -C_{31} & -C_{32} & C_{31} + C_{32} + C_{33} \end{pmatrix} \begin{pmatrix} \dot{V}_1 \\ \dot{V}_2 \\ \dot{V}_3 \end{pmatrix} = \begin{pmatrix} \frac{V_{dd} - V_1}{R_0} - \frac{V_1}{R_{VO_21}} \\ \frac{V_{dd} - V_2}{R_0} - \frac{V_2}{R_{VO_22}} \\ \frac{V_{dd} - V_3}{R_0} - \frac{V_3}{R_{VO_23}} \end{pmatrix}$$

In the application presented in this work, the coupling capacitors value are all same: $C_{ij} = C_o (i \neq j)$.

Hence, the natural frequency of the i -th oscillator can be altered by adjusting C_{ii} . Solving a coupled system of n oscillators can be accomplished in the following:

$$V_i(t = 0) = V_{i0}, R_{VO_2i} = R_{off}$$

for $0 : dt : T$

$$\text{compute } b_i = \frac{V_{dd} - V_i}{R_0} - \frac{V_i}{R_{VO_2i}}$$

$$\text{compute } a_{ii} = \sum_j C_{ij}, a_{ij} = -C_{ij}$$

$$dV = \text{linsolve}(b, a) \times dt$$

$$V(t + dt) = V(t) + dV$$

for $i = 1 : n$

$$\text{if } (R_{VO_2i} = R_{off}) \& (V_i(t + dt)) \geq V_{th} + C_{var1} * \text{rand}$$

$$R_{VO_2i} = R_{on};$$

$$\text{if } (R_{VO_2i} = R_{on}) \& (V_i(t + dt)) \leq V_{hold} + C_{var2} * \text{rand}$$

$$R_{VO_2i} = R_{off};$$

end

end

We simulated the coupling process of 2 and 3 oscillators using MATLAB and investigated the impact of a broad range of parameters. As for the device model, $V_{th} = 3.8$ V, $V_{hold} = 1.6$ V, $R_{off} = 20$ kohm, $R_{on} = 300$ ohm.

For two-oscillator coupling, We set $V_{dd} = 8$ V, $R_L = 10$ kohm, $C_o = 300$ pF. We set one sensing capacitor is 400 pF, so that the natural frequency of a single oscillator is 402 kHz. Then we change the other oscillator's sensing capacitor to observe the two oscillators' coupling situation. The result is shown in Supplementary Fig 21.

The results depicted in the figure reveal that only when the other oscillator sensing capacitor is in 310

pF- 500 pF, the two oscillators can be synchronized and lock phase difference. The difference in natural frequency is 117 kHz (310pF) and 80kHz (500pF).

When increasing the coupling capacitance, the tolerance of the natural frequency difference will also increase. As shown in Supplementary Fig 22(a), when the coupling capacitance is increased to 500 pF, the natural frequency range is from 310 kHz (510 pF) to 536 kHz (300 pF); when the coupling capacitance is increased to 1 nF, the natural frequency range is from 293 kHz (500 pF) to 555 kHz (290 pF). However, due to the real physical devices having device-to-device variation, the coupling capacitance cannot be too large, otherwise the synchronization will be destroyed. For example, when device 1 and device 2 have different V_{th} , $V_{th1} = 3.8V$, $V_{th2} = 3.7V$, and the coupling capacitance exceeds 1.6 nF, the two devices cannot synchronize as shown in Supplementary Fig 22(b)

For three-oscillator coupling, the parasitic capacitance is set at 200 pF. As shown in Supplementary Fig 23, With the two devices' parallel capacitors fixed at 0.4nF, varying the third capacitance value revealed a valid coupling range in 0.3nF-0.7nF. Furthermore, when two devices' parallel capacitors were set at 0.4nF and one device's parallel capacitor was set at 0.6nF, we modified the coupling capacitor value to observe potential coupling. As illustrated in Supplementary Fig 24 (a) and (b), we observed that coupling persisted until the value increased to 700 pF and 800 pF in case of mode 001 and 011 respectively.

Device-to-device and cycle-to-cycle variability in VO_2 oscillators also have an impact on coupling. Taking the variation of V_{th} as an example, we can configure the coupling state to 000, 001, 011 and 111 mode, setting the V_{th} of two devices simultaneously to 3.8 and manipulating the V_{th} of the third device. The result in Supplementary Fig 25 shows that coupling is unattainable when V_{th} is below 3.5 at 001 mode, but it remains feasible when V_{th} exceeds 4.1V at all case. This outcome substantiates that coupling is established within the range of device fluctuations ($3.5 < V_{th} < 4.1V$).

Cycle-to-cycle variation will influence the phase resolution post coupling. The phase jitter induced by V_{th} fluctuate during the oscillation represents the lower limit of phase resolution. As shown in Supplementary Fig 26, we have evaluated the impact of cycle-to-cycle variation in different modes. When the cycle-to-cycle variation increases from 0.01 to 0.11, the maximum value of the phase difference is less than 0.35° . Considering that the cycle-to-cycle variation of the device is within 0.01 (Supplementary Fig 4), The impact of this part is almost negligible.

The variations of other parameters from device to device or cycle to cycle can also be analyzed in a similar way using the formulas and framework we proposed here.

Reviewer #2 (Remarks to the Author):

Authors revised the abstract and it is in acceptable form now. However, overall, the paper is not well-written. I had a comment on wrong terminology of calling neurons memristive and the authors did not understand my comment and instead wrote a long response on history of memristors which was totally unnecessary as what memristor is well-known to all the reviewers. I think this is just an oscillatory neuron, so terminology needs to be corrected. Fig. R1 suggests there were issues with the fabrication. Experiment setup is a breadboard (Sup Fig 11). Doesn't it suffer from parasitics? It is hard to understand why the authors could not design a PCB to integrate everything on it.

Our response:

We would like to thank the reviewer for pointing out the deficiency in the text. We apologize for any misunderstanding regarding the terminology of calling neurons memristive. Our initial intention was to state that the device employed for implementing the oscillatory neuron itself is memristive. However, we acknowledge that this may have led to some misunderstandings. Therefore, we have revised the manuscript to avoid referring to memristive when using the word neuron, and only emphasize the oscillating characteristic

Regarding the fabrication issues suggested by Fig. R1, we should clarify that the stains visible in the photos are not present in the as-fabricated sample, but are later contamination caused by improper storage of the sample. In addition, these are surface contaminations and do not have

a significant impact on device performance.

For the use of a breadboard in the experiment setup (Sup Fig 11), in our experiments, the breadboard is solely employed for implementing only 3 coupling capacitors (300pF) and 3 series resistors (8.2K Ω). These components are passive in nature. Additionally, the coupling capacitor we use is 300pF, while the typical parasitic capacitance of a standard breadboard ranges from 1-10pF. For instance, the maximum parasitic capacitance reported in {López, Carlos Sánchez. “A 1.7 MHz Chua's circuit using VMs and CF+s.” *Revista Mexicana De Fisica* 58 (2012): 86-93.} is only 2.7pF. At the same time, we investigated the effects of changes in coupling capacitance on coupled oscillations in the revised **Supplementary Note 2**. It is evident that these parasitic effects' influence on our experimental demonstration is not expected to be significant.

We would like to thank the reviewer once again for the valuable feedback, and we hope that the reviewer will be satisfied with the revised version.

Reviewer #3 (Remarks to the Author):

Thanks Authors for the the great job and the efforts you did for the revision part. The comments were well addressed and the paper is now more clear and better explained.

Our response:

We would like to thank the reviewer for the positive feedback on the revised manuscript. We greatly appreciate the reviewer's recognition of our efforts in improving the manuscript.

REVIEWERS' COMMENTS

Reviewer #1 (Remarks to the Author):

The authors have addressed my questions, yet two aspects still lack clarity:

1. derivation of capacitive coupling values remains nontrivial, depending on the type of sensor signals.
2. There may be a fundamental scalability challenge associated with this sensing approach. With n oscillators, accommodating 2^n states (modes as reported in paper) results in the division of the 360-degree phase plot by n oscillators. The minimum phase difference required to distinguish between two oscillators in all 2^n states can be calculated as $360/2^n$. For example, in the case of 3 oscillators, the minimum phase difference between two oscillators in all 8 states would be 45 degrees, aligning with the authors' findings in Case "111." Consequently, even with 4 oscillators, distinguishing sensor signals prompting oscillators with $360/16=22.5$ degrees could prove challenging, and even more so with the increasing number of oscillators.

Reviewer #2 (Remarks to the Author):

The authors responded to my comments. I do not have further comments.

MS No: NCOMMS-23-18953B

Title: High-order sensory processing nanocircuit based on coupled VO2 oscillators

Reviewer #1 (Remarks to the Author):

1 derivation of capacitive coupling values remains nontrivial, depending on the type of sensor signals.

Our response: We would like to thank the reviewer for raising this constructive question.

In Supplementary Note 2, We have presented the analysis and mathematical model for coupling. It is evident that the solution of capacitive coupling values involves solving differential equations, making direct mathematical derivation a challenging task. Therefore, we have provided a numerical solution method, described in Supplementary Note 2 to address this complexity. Concurrently, we validate the correctness and effectiveness of the solver by exemplifying its application in the two-oscillator coupling and three-oscillator coupling scenarios explored in this study. It's worth noting that this method is inherently scalable and readily adaptable to cases involving n-oscillator coupling.

For the capacitive coupling employed in our work, various sensor signals first undergo transformation into distinct parallel capacitances through the sensor. Subsequently, these capacitances are functioned in the same manner. Therefore, our coupled oscillators demonstrate insensitivity to the type of sensing signal. Furthermore, we have thoroughly analyzed and elucidated the effective capacitance of the parallel capacitor in Supplementary Note 2, along with supporting information in Supplementary Fig 21 and Supplementary Fig 23, which describes the range of capacitor values that are valid in our configuration.

2 There may be a fundamental scalability challenge associated with this sensing approach. With n oscillators, accommodating 2^n states (modes as reported in paper) results in the division of the 360-degree phase plot by n oscillators. The minimum phase difference required to distinguish between two oscillators in all 2^n states can be calculated as $360/2^n$. For example, in the case of 3 oscillators, the minimum phase difference between two oscillators in all 8 states would be 45 degrees, aligning with the authors' findings in Case "111." Consequently, even with 4 oscillators, distinguishing sensor signals prompting oscillators with $360/16=22.5$ degrees could prove challenging, and even more so with the increasing number of oscillators.

Our response: We would like to thank the reviewer for highlighting the critical issues of the minimum phase difference.

As the reviewer rightly pointed out, resolving finer-grained phase differences is challenging. Overcoming this challenge necessitates the implementation of more robust subsequent circuits for post-processing, such as phase detectors with higher reference frequencies. Phase detection circuits can typically achieve sub-degree or higher level precision, and have large number of applications in various fields such as digital communication systems, radar detection, and medical imaging. However, enhancing performance often involves increased power consumption and circuit area. Therefore, in specific coupled oscillation applications, it is essential to strike a trade-off between accuracy and cost.

Additionally, the minimal phase difference is determined by the cycle-to-cycle variation inherent in the VO₂ based oscillator. We conduct a detailed analysis of this effect based on coupled oscillation model and solver in Supplementary Note 2. Considering that the cycle-to-cycle variation of the device falls within the range of 0.01 (Supplementary Fig 4), we configured the cycle-to-cycle variation coefficient to be 0.01. Under these conditions, the minimum value of the phase difference is found to be within 0.046° which corresponds to a coupling of approximately 88 oscillators. When the cycle-to-cycle variation increases from 0.01 to 0.11, the maximum value of the phase difference remains below 0.35° which corresponds to a coupling of approximately 32 oscillators.

Reviewer #2 (Remarks to the Author):

The authors responded to my comments. I do not have further comments.

Our response: We are delighted that the reviewer is satisfied with our manuscript and we greatly appreciate the reviewer's recognition of our efforts in improving the manuscript.